# Dynamics of striatal action selection and reinforcement learning

Jack W Lindsey[1], Jeffrey Markowitz[2], Winthrop F Gillis[3], Sandeep R Datta[3], Ashok Litwin-Kumar[1]*

[1]Kavli Institute for Brain Science, Columbia University, New York, United States; [2]Wallace H. Coulter Department of Biomedical Engineering, Georgia Institute of Technology and Emory University, Atlanta, United States; [3]Department of Neurobiology, Harvard Medical School, Boston, United States

## eLife Assessment

The authors present a biologically plausible framework for action selection and learning in the striatum that is a **fundamental** advance in our understanding of possible neural implementations of reinforcement learning in the basal ganglia. They provide **compelling** evidence that their model can reconcile realistic neural plasticity rules with the distinct functional roles of the direct and indirect spiny projection neurons of the striatum, recapitulating experimental findings regarding the activity profiles of these distinct neural populations and explaining a key aspect of striatal function.

*For correspondence:
a.litwin-kumar@columbia.edu

**Abstract** Spiny projection neurons (SPNs) in dorsal striatum are often proposed as a locus of reinforcement learning in the basal ganglia. Here, we identify and resolve a fundamental inconsistency between striatal reinforcement learning models and known SPN synaptic plasticity rules. Direct-pathway (dSPN) and indirect-pathway (iSPN) neurons, which promote and suppress actions, respectively, exhibit synaptic plasticity that reinforces activity associated with elevated or suppressed dopamine release. We show that iSPN plasticity prevents successful learning, as it reinforces activity patterns associated with negative outcomes. However, this pathological behavior is reversed if functionally opponent dSPNs and iSPNs, which promote and suppress the current behavior, are simultaneously activated by efferent input following action selection. This prediction is supported by striatal recordings and contrasts with prior models of SPN representations. In our model, learning and action selection signals can be multiplexed without interference, enabling learning algorithms beyond those of standard temporal difference models.

## Introduction

Numerous studies have proposed that the basal ganglia is a reinforcement learning system (*Joel et al., 2002*; *Niv, 2009*; *Ito and Doya, 2011*). Reinforcement learning algorithms use experienced and predicted rewards to learn to predict the expected future reward associated with an organism's current state, and the action to select, in order to maximize this reward (*Sutton and Barto, 2018*). Spiny projection neurons (SPNs) in the striatum are well-positioned to take part in such an algorithm, as they receive diverse contextual information from the cerebral cortex and are involved in both action selection (in dorsal striatum; *Packard and Knowlton, 2002*; *Seo et al., 2012*; *Balleine et al., 2007*) and value prediction (in ventral striatum; *Cardinal et al., 2002*; *Montague et al., 1996*; *O'Doherty et al., 2004*). Moreover, plasticity of SPN input synapses is modulated by midbrain dopamine release (*Wickens et al., 1996*; *Calabresi et al., 2000*; *Contreras-Vidal and Schultz, 1999*). A variety of studies support the view that this dopamine release reflects reward prediction error (*Schultz et al.,*

*1997*; *Montague et al., 1996*; *Houk and Adams, 1995*), which in many reinforcement learning algorithms is the key quantity used to modulate learning (*Sutton and Barto, 2018*; *Niv, 2009*).

Despite these links, several aspects of striatal physiology are difficult to reconcile with reinforcement learning models. SPNs are classified in two main types – direct-pathway (dSPNs) and indirect-pathway (iSPNs). These two classes of SPNs exert opponent effects on action based on perturbation data (*Kravitz et al., 2010*; *Freeze et al., 2013*; *Lee and Sabatini, 2021*), but also exhibit highly correlated activity (*Cui et al., 2013*). Moreover, dSPNs and iSPNs express different dopamine receptors (D1- and D2-type) and thus undergo synaptic plasticity according to different rules. In particular, dSPN inputs are potentiated when coincident pre- and post-synaptic activity is followed by above-baseline dopamine activity, while iSPN inputs are potentiated when coincident pre- and post-synaptic activity is followed by dopamine suppression (*Shen et al., 2008*; *Frank, 2005*; *Iino et al., 2020*).

Prior studies have proposed that dSPNs learn from positive reinforcement to promote actions, and iSPNs learn from negative reinforcement to suppress actions (*Cruz et al., 2022*; *Collins and Frank, 2014*; *Jaskir and Frank, 2023*; *Varin et al., 2023*; *Mikhael and Bogacz, 2016*; *Dunovan et al., 2019*). However, we will show that a straightforward implementation of such a model fails to yield a functional reinforcement learning algorithm, as the iSPN learning rule assigns blame for negative outcomes to the wrong actions. Correct learning in this scenario requires a mechanism to selectively update corticostriatal weights corresponding to the chosen action, which is absent in prior models (see Discussion).

In this work, we begin by rectifying this inconsistency between standard reinforcement learning models of the striatum and known SPN plasticity rules. The iSPN learning rule reported in the literature reinforces patterns of iSPN activity that are associated with dopamine suppression, increasing the likelihood of repeating decisions that previously led to negative outcomes. We show that this pathological behavior is reversed if, after action selection, opponent dSPNs and iSPNs receive correlated efferent input encoding the animal's selected action. A central contribution of our model is a decomposition of SPN activity into separate modes for action selection and for learning, the latter driven by this efferent input. This decomposition provides an explanation for the apparent paradox that the activities of dSPNs and iSPNs are positively correlated despite their opponent causal functions (*Cui et al., 2013*), and provides a solution to the problem of multiplexing signals related to behavioral execution and learning. The model also makes predictions about the time course of SPN activity,

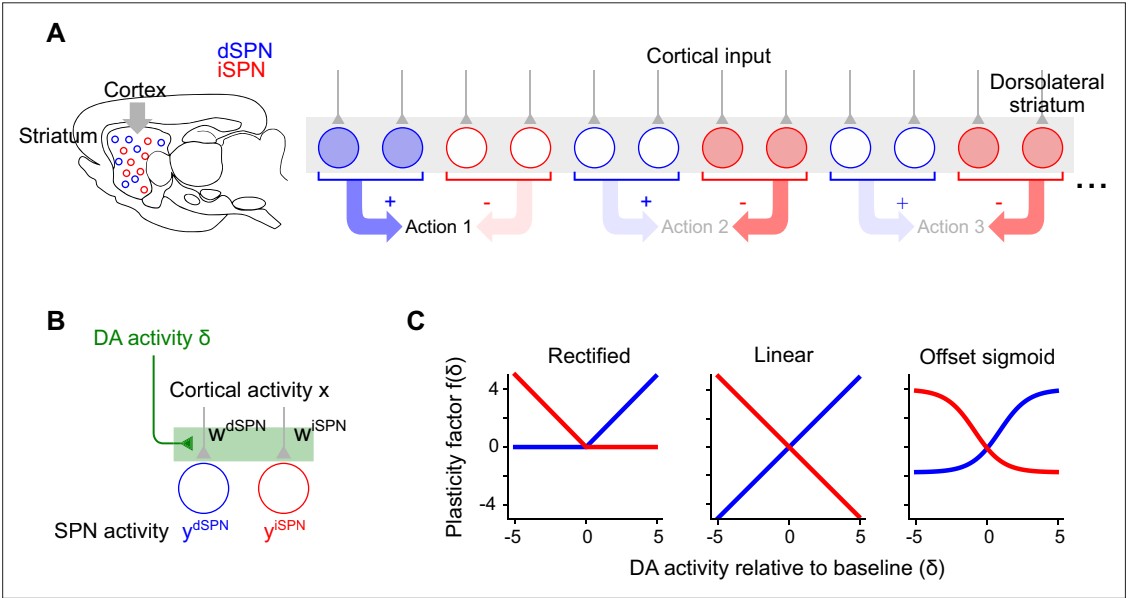

**Figure 1.** Corticostriatal action selection circuits and plasticity rules. (**A**) Left, diagram of cortical inputs to striatal populations. Right, illustration of action selection architecture. Populations of dSPNs (blue) and iSPNs (red) in dorsolateral striatum (DLS) are responsible for promoting and suppressing specific actions, respectively. Active neurons (shaded circles) illustrate a pattern of activity consistent with typical models of striatal action selection, in which dSPNs that promote a chosen action and iSPNs that suppress other actions are active. (**B**) Illustration of three-factor plasticity rules at spiny projection neuron (SPN) input synapses, in which adjustments to corticostriatal synaptic weights depend on pre-synaptic cortical activity, SPN activity, and dopamine release. (**C**) Illustration of different models of the dopamine-dependent factor $f(\delta)$ in dSPN (blue) and iSPN (red) plasticity rules.

including that dSPNs and iSPNs that are responsible for regulating the same behavior (promoting and suppressing it, respectively) should be co-active following action selection. This somewhat counterintuitive prediction contrasts with prior proposals that dSPNs that promote an action are co-active with iSPNs that suppress different actions (*Mink, 1996*; *Redgrave et al., 1999*). We find support for this prediction in experimental recordings of dSPNs and iSPNs during spontaneous behavior.

Next, we show that the nonuniformity of dSPN and iSPN plasticity rules enables more sophisticated learning algorithms than can be achieved in models with a single plasticity rule. In particular, it enables the striatum to implement so-called *off-policy* reinforcement learning algorithms, in which the corticostriatal pathway learns from the the outcomes of actions that are driven by other neural pathways. Off-policy algorithms are commonly used in state-of-the-art machine learning models, as they dramatically improve learning efficiency by facilitating learning from expert demonstrations, mixture-of-experts models, and replayed experiences (*Arulkumaran et al., 2017*). Following the implications of this model further, we show that off-policy algorithms require a dopaminergic signal in dorsal striatum that combines classic state-based reward prediction error with a form of action prediction error. We confirm a key signature of this prediction in recent dopamine data collected from dorsolateral striatum (DLS) during spontaneous behavior.

## Results

In line with previous experimental (*Wickens et al., 1996*; *Calabresi et al., 2000*; *Contreras-Vidal and Schultz, 1999*) and modeling (*Sutton and Barto, 2018*; *Niv, 2009*) studies, we model plasticity of corticostriatal synapses using a three-factor learning rule, dependent on coincident presynaptic activity, post-synaptic activity, and dopamine release (*Figure 1A, B*). Concretely, we model plasticity of the weight $w$ of a synapse from a cortical neuron with activity $x$ onto a dSPN or iSPN with activity $y$ as

$$\Delta w^{\mathrm{dSPN}} = f^{\mathrm{dSPN}}(\delta) \cdot y^{\mathrm{dSPN}} \cdot x, \tag{1}$$

$$\Delta w^{\mathrm{iSPN}} = f^{\mathrm{iSPN}}(\delta) \cdot y^{\mathrm{iSPN}} \cdot x, \tag{2}$$

where $\delta$ represents dopamine release relative to baseline, and the functions $f^{\mathrm{dSPN}}(\delta)$ and $f^{\mathrm{iSPN}}(\delta)$ model the dependence of the two plasticity rules on dopamine concentration.

For dSPNs, the propensity of input synapses to potentiate increases with increasing dopamine concentration, while for iSPNs the opposite is true. This observation is corroborated by converging evidence from observations of dendritic spine volume, intracellular PKA measurements, and spike-timing-dependent plasticity protocols (*Shen et al., 2008*; *Gurney et al., 2015*; *Iino et al., 2020*; *Lee et al., 2021*). For the three-factor plasticity rule above, these findings imply that $f^{\mathrm{dSPN}}$ is an increasing function of $\delta$ while $f^{\mathrm{iSPN}}$ is a decreasing function. Prior modeling studies have proposed specific plasticity rules that correspond to different choices of $f^{\mathrm{dSPN}}$ and $f^{\mathrm{iSPN}}$, some examples of which are shown in *Figure 1C*.

### iSPN plasticity rule impedes successful reinforcement learning

Prior work has proposed that dSPNs activate when actions are performed and iSPNs activate when actions are suppressed (*Figure 1A*). When an animal selects among multiple actions, subpopulations of dSPNs are thought to promote the selected action, while other subpopulations of iSPNs inhibit the unchosen actions (*Mink, 1996*; *Redgrave et al., 1999*). We refer to this general description as the 'canonical action selection model' of SPN activity and show that this model, when combined with the plasticity rules above, fails to produce a functional reinforcement learning algorithm. This failure is specifically due to the iSPN plasticity rule. Later, we also show that the SPN representation predicted by the canonical action selection model is inconsistent with recordings of identified dSPNs and iSPNs. We begin by analyzing a toy model of an action selection task with two actions, one of which is rewarded. In the model, the probability of selecting an action is increased when the dSPN corresponding to that action is active and decreased when the corresponding iSPN is active. After an action is taken, dopamine activity reports the reward prediction error, increasing when reward is obtained and decreasing when it is not.

It is easy to see that the dSPN plasticity rule in *Equation 1* is consistent with successful reinforcement learning (*Figure 2A*). Suppose action 1 is selected, leading to reward (*Figure 2A*, center). The

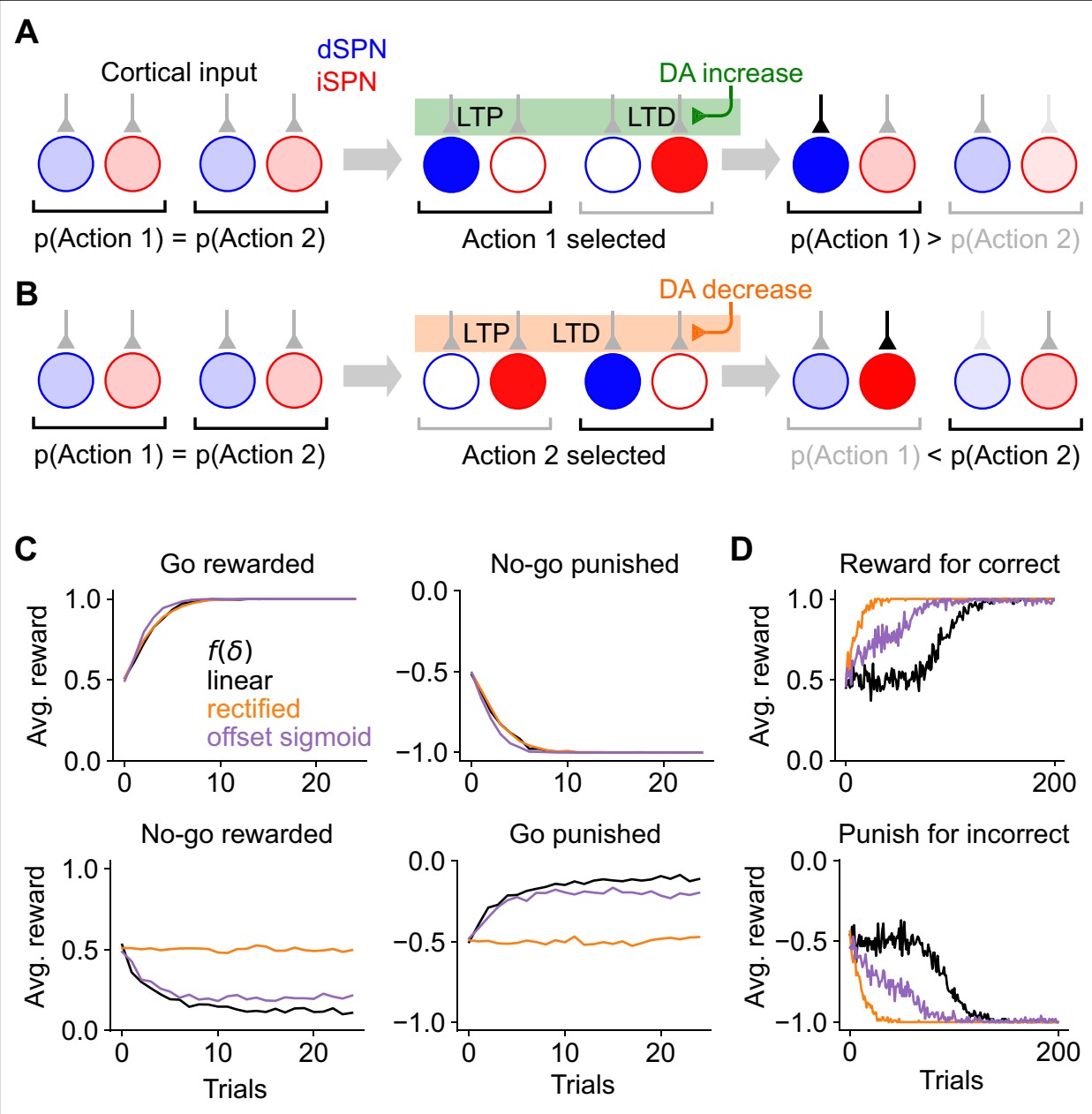

**Figure 2.** Consequences of the canonical action selection model of spiny projection neuron (SPN) activity. (**A**) Example in which dSPN plasticity produces correct learning. Left: cortical inputs to the dSPN and iSPN are equal prior to learning. Shading of corticostriatal connections indicates synaptic weight, and shading of blue and red circles denotes dSPN/iSPN activity. Middle: action 1 is selected, corresponding to elevated activity in the dSPN that promotes action 1 and the iSPN that suppresses action 2. In this example, action 1 leads to reward and increased DA activity. This potentiates the input synapse to the action 1-promoting dSPN and (depending on the learning rule, see *Figure 1*) depresses the input to the action 2-suppressing iSPN. Right: in a subsequent trial, cortical input to the action 1-promoting dSPN is stronger, increasing the likelihood of selecting action 1. Here, the dSPN-mediated effect of increasing action 1's probability overcomes the iSPN-mediated effect of decreasing action 2's probability. (**B**) Example in which iSPN plasticity produces incorrect learning. Same as A, but in a scenario in which action 2 is selected leading to punishment and a corresponding decrease in DA activity. As a result, the input synapse to the action 2-promoting dSPN is (depending on the learning rule) depressed, and the input to the action 1-suppressing iSPN is potentiated. On a subsequent trial, the probability of selecting action 2 rather than action 1 is greater, despite action 2 being punished. Note that the dSPN input corresponding to action 2 is (potentially) weakened, which correctly decreases the probability of selecting action 2, but this effect is not sufficient to overcome the strengthened action 1 iSPN activity. (**C**) Performance of a simulated striatal reinforcement learning system in go/no-go tasks with different reward contingencies. (**D**) Same as C, but for action selection tasks with two cortical input states, two available actions, and one correct action per state, under different reward protocols.

The online version of this article includes the following figure supplement(s) for figure 2:

**Figure supplement 1.** Go/no-go task.

resulting dopamine increase potentiates inputs to the action 1 dSPN from cortical neurons that are active during the task, making action 1 more likely to be selected in the future (*Figure 2A*, right).

At first glance, it may seem that a similar logic would apply to iSPNs, since their suppressive effect on behavior and reversed dependence on dopamine concentration are both opposite to dSPNs. However, a more careful examination reveals that the iSPN plasticity rule in *Equation 2* does not promote successful learning. In the canonical action selection model, dSPNs promoting a selected action and iSPNs inhibiting unselected actions are active. If a negative outcome is encountered leading to a dopamine decrease, *Equation 2* predicts that inputs to iSPNs corresponding to unselected actions are strengthened (LTP in *Figure 2B*, center). This makes the action that led to the negative outcome *more* rather than less likely to be taken when the same cortical inputs are active in the future (*Figure 2B*, right). More generally, the model demonstrates that, while the plasticity rule of *Equation 1* correctly reinforces dSPN activity patterns that lead to positive outcomes, *Equation 2* incorrectly reinforces iSPN activity patterns that lead to negative outcomes. The function of iSPNs in inhibiting action does not change the fact that such reinforcement is undesirable.

We note that, depending on the learning rule (*Figure 1C*), inputs to dSPNs that promote the selected action may be weakened (LTD in *Figure 2B*, left), which correctly disincentivizes the action that led to a negative outcome. However, this dSPN effect competes with the pathological behavior of the iSPNs and is often unable to overcome it. We also note that, if dopamine increases lead to depression of iSPN inputs (*Figure 1A*, center, right), positive outcomes will lead to actions that were correctly being inhibited by iSPNs to be less inhibited in the future. Thus, both positive and negative outcomes may cause incorrect iSPN learning. Some sources suggest that while dopamine suppression increases D2 receptor activation, dopamine increase has little effect on D2 receptors (*Dreyer et al., 2010*), corresponding to the rectified model of $f(\delta)$ (*Figure 1C*, left). In this case, pathological iSPN plasticity behavior still manifests when dopamine activity is suppressed (as in the examples of *Figure 2B*).

We simulated learning of multiple tasks with the three-factor plasticity rules above, with dopamine activity modeled as reward prediction error obtained using a temporal difference (TD) learning rule. In a go/no-go task with one cue in which the 'go' action is rewarded (*Figure 2—figure supplement 1*), the system learns the wrong behavior when negative performance feedback is provided on no-go trials, and thus iSPN plasticity is the main driver of learning (*Figure 2C*). We also simulated a two-alternative forced choice task in which there are two cues (corresponding to different cortical input patterns), each with a corresponding target action. When performance feedback consists of rewards for correct actions, the system learns the task, as dSPNs primarily drive the learning. However, when instead performance feedback consists of giving punishments for incorrect actions, the system does not learn the task, as iSPNs primarily drive the learning (*Figure 2D*). We note that, in principle, this problem could be avoided if the learning rate of iSPNs were very small compared to that of dSPNs, ensuring that reinforcement learning is always primarily driven by the dSPN pathway (leaving iSPNs to potentially perform a different function). However, this alternative would be inconsistent with prior studies indicating a significant role for the indirect pathway in reinforcement learning (*Peak et al., 2020*; *Lee and Sabatini, 2021*). The model we introduce below makes use of contributions to learning from both pathways.

## Efferent activity in SPNs enables successful reinforcement learning

We have shown that the canonical action selection model, when paired with *Equations 1 and 2*, produces incorrect learning. What pattern of SPN activity would produce correct learning? In the model, the probability of selecting an action is determined by the 'difference mode' $y^{\text{dSPN}} - y^{\text{iSPN}}$, where $y^{\text{dSPN}}$ and $y^{\text{iSPN}}$ are the activities of dSPN and iSPN neurons associated with that action. We analyzed how the plasticity rule of *Equations 1 and 2* determines changes to this difference mode. In the simplest case in which the SPN firing rate is a linear function of cortical input (i.e., $y^{\text{d/iSPN}} = \mathbf{w}^{\text{d/iSPN}} \cdot \mathbf{x}$) and plasticity's dependence on dopamine concentration is also linear (i.e., $f^{\text{d/iSPN}}(\delta) \propto \pm\delta$; *Figure 1C*, center), the change in the probability of selecting an action due to learning is

$$
\begin{aligned}
\Delta(y^{\text{dSPN}} - y^{\text{iSPN}}) &= \Delta\mathbf{w}^{\text{dSPN}} \cdot \mathbf{x} - \Delta\mathbf{w}^{\text{iSPN}} \cdot \mathbf{x} \\
&\propto \delta y^{\text{dSPN}}(\mathbf{x} \cdot \mathbf{x}) - (-\delta)y^{\text{iSPN}}(\mathbf{x} \cdot \mathbf{x}) \\
&\propto \delta(y^{\text{dSPN}} + y^{\text{iSPN}}).
\end{aligned}
\tag{3}
$$

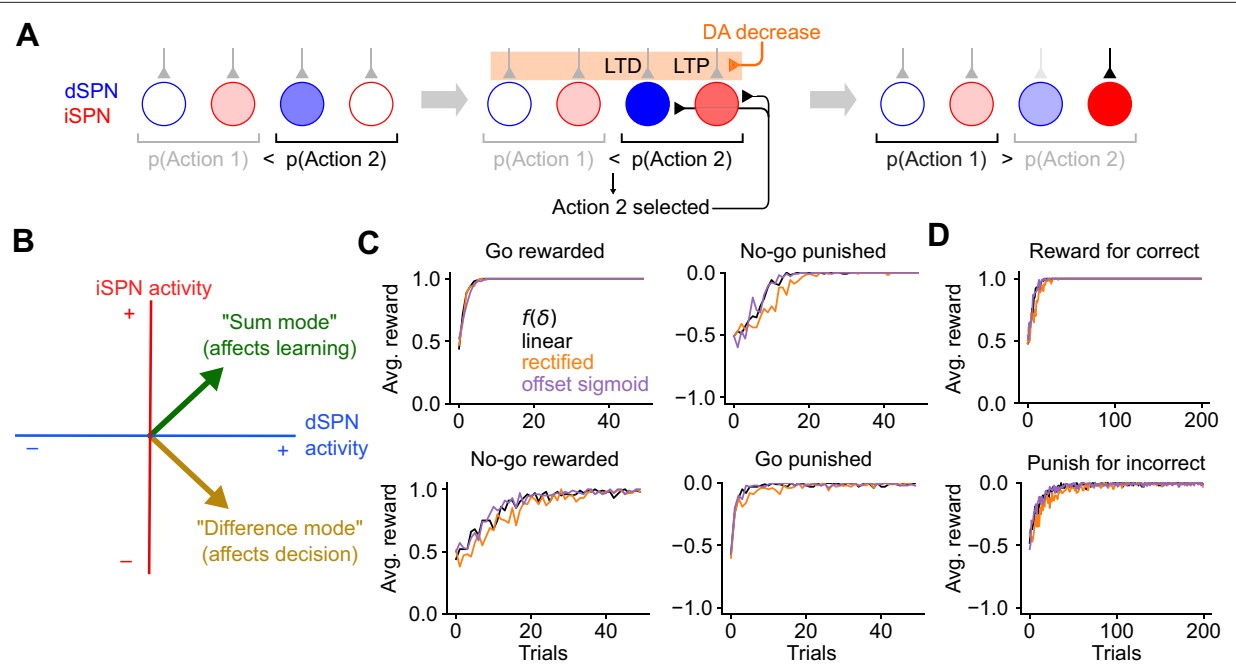

**Figure 3.** The efference model of spiny projection neuron (SPN) activity. (**A**) Illustration of the efference model in an action selection task. Left: feedforward SPN activity driven by cortical inputs. Center: once action 2 is selected, efferent inputs excite the dSPN and iSPN responsible for promoting and suppressing action 2. Efferent activity is combined with feedforward activity, such that the action 2-associated dSPNs and iSPNs are both more active than the action 1 dSPNs and iSPNs, but the relative dSPN and iSPN activity for each action remains unchanged. This produces strong LTD and LTP in the action 2-associated dSPNs and iSPNs upon a reduction in dopamine activity. Right: in a subsequent trial, this plasticity correctly reduces the likelihood of selecting action 2. (**B**) The activity levels of the dSPN and iSPN populations that promote and suppress a given action can be plotted in a two-dimensional space. The difference mode influences the probabiility of taking that action, while the sum mode drives future changes to activity in the difference mode via plasticity. Efferent activity excites the sum mode. (**C**) Performance of a striatal RL system using the efference model on the tasks of *Figure 2C*. (**D**) Performance of a striatal RL system using the efference model on the tasks of *Figure 2D*.

The online version of this article includes the following figure supplement(s) for figure 3:

**Figure supplement 1.** Comparison of canonical action selection and efference models with a distributed action code.

Changes to the 'difference mode' $y^{\text{dSPN}} - y^{\text{iSPN}}$ are therefore driven by the 'sum mode' $y^{\text{dSPN}} + y^{\text{iSPN}}$. This implies that the activity pattern that leads to correct learning about an action's outcome is different from the activity pattern that selects the action. To promote or inhibit, respectively, an action that leads to a dopamine increase or decrease, this analysis predicts that both dSPNs that promote and iSPNs that inhibit the action should be co-active. A more general argument applies for other learning rules and firing rate nonlinearities: as long as $y^{\text{d/iSPN}}$ is an increasing function of total input current, $f^{\text{dSPN}}(\delta)$ has positive slope, and $f^{\text{iSPN}}(\delta)$ has negative slope, changes in difference mode activity will be positively correlated with sum mode activity (see Appendix).

The key insight of the above argument is that the pattern of SPN activity needed for learning involves simultaneous excitation of dSPNs that promote the current behavior and iSPNs that inhibit it. This differs from the pattern of activity needed to drive selection of that behavior in the first place. We therefore propose a model in which SPN activity contains a substantial *efferent* component that follows action selection and promotes learning, but has no causal impact on behavior. In the model, feedforward corticostriatal inputs initially produce SPN activity whose difference mode causally influences action selection, consistent with the canonical model (*Figure 3A*, left). When an action is performed, both dSPNs and iSPNs responsible for promoting or inhibiting that action receive efferent excitatory input, producing sum mode activity. Following this step, SPN activity reflects both contributions (*Figure 3A*, center). The presence of sum mode activity leads to correct synaptic plasticity and learning (*Figure 3A*, right). Unlike the canonical action selection model (*Figure 1A*), this model thus predicts an SPN representation in which, after an action is selected, the most highly active neurons are those responsible for regulating that behavior and not other behaviors.

In SPN activity space, the sum and difference modes are orthogonal to one another. This orthogonality has two consequences. First, it implies that encoding the action in the difference mode (as in the canonical action selection model) produces synaptic weight changes that do not promote learning, consistent with the competing effects of dSPN and iSPN plasticity that we previously described. Second, it implies that adding efferent activity along the sum mode, which produces correct learning, has no effect on action selection. The model thus provides a solution to the problem of interference between 'forward pass' (action selection) and 'backward pass' (learning) activity, a common issue in models of biologically plausible learning algorithms (see Discussion).

In simulations, we confirm that unlike the canonical action selection model, this efference model solves go/no-go (*Figure 3C*) and action selection (*Figure 3D*) tasks regardless of the reward protocol. Although the derivation above assumes linear SPN responses and linear dependences of plasticity on dopamine concentration, our model enables successful learning even using a nonlinear model of SPN responses and a variety of plasticity rules (*Figure 3C, D*; see Appendix for a derivation that explains this general success). Finally, we also confirmed that our results apply to cases in which actions are associated with distributed modes of dSPN and iSPN activity, and with a larger action space (*Figure 3— figure supplement 1*). This success arises from the ability to form orthogonal subspaces for action selection and learning in this distributed setting. Although we describe the qualitative behavior of our model using discrete action spaces for illustrative purposes, we expect distributed representations to be more faithful to neural recordings.

## Temporal dynamics of the efference model

We simulated a two-alternative forced choice task using a firing rate model of SPN activity. This allowed us to directly visualize dynamics in the sum and difference modes and verify that the efference model prevents interference between them. In each trial of the forced choice task, one of two stimuli is presented and one of two actions is subsequently selected (*Figure 4A*, top row). The selected action is determined by the difference mode activity of action-encoding SPNs during the first half of the stimulus presentation period. The sum mode is activated by efferent input during the second half of this period. Reward is obtained if the correct action is selected in a trial, and each stimulus has a different corresponding correct action. Plasticity of cortical weights encoding stimulus identity onto SPNs is governed by *Equations 1 and 2*.

The model learned the correct policy in about 10 trials. Early in learning, difference mode activity is small and primarily driven by noise, leading to random action selection (*Figure 4B*). However, sum mode activity is strongly driven after an action is selected (*Figure 4B*, bottom). As learning progresses, the magnitude of the difference mode activity evoked by the stimulus increases (*Figure 4B*, third row). Late in learning, dSPN and iSPN firing rates are more separable during stimulus presentation, leading to correct action selection (*Figure 4C*, second row). Both difference and sum mode activity is evident late in learning, with the former leading the latter (*Figure 4C*, bottom two rows).

Throughout the learning process, difference and sum mode activity for the two actions are separable and non-interfering, even when both are present simultaneously. As a result, action selection is not disrupted by efferent feedback. We conclude that the efference model multiplexes action selection and learning signals without separate learning phases or gated plasticity rules. While we illustrated this in a task with sequential trials for visualization purposes, this non-interference enables learning based on delayed reward and efferent feedback from past actions even as the selection of subsequent actions unfolds.

## Efference model predicts properties of SPN activity

Thus far, we have provided theoretical arguments and model simulations that suggest that simultaneous efferent input to opponent dSPNs and iSPNs is necessary for reinforcement learning, given known plasticity rules. We next sought to test this prediction in neural data. We predict these dynamics to be particularly important in scenarios where the action space is large and actions are selected continuously, without a clear trial structure. We therefore used data from a recent study which recorded bulk and cellular dSPN and iSPN activity in spontaneously behaving mice (*Figure 5A*; *Markowitz et al., 2018*). As no explicit rewards or task structure were provided during recording sessions, we adopted a modeling approach that makes minimal assumptions about the inputs to SPNs besides the core prediction of efferent activity. Specifically, we used a network model in which (1)

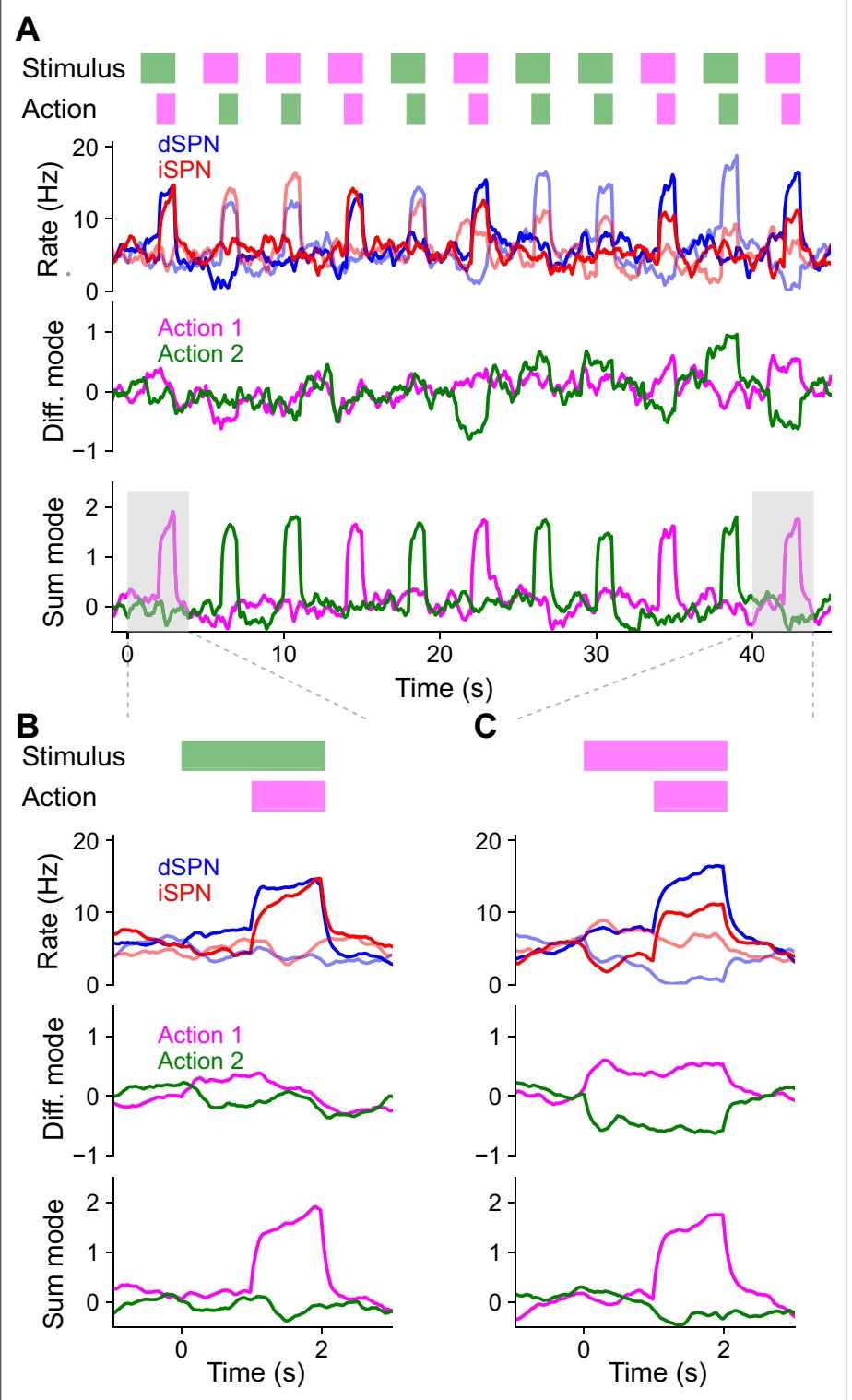

**Figure 4.** Temporal dynamics of the efference model in a two-alternative forced choice task. (**A**) Top row: in each trial, either stimulus 1 (magenta) or stimulus 2 (green) is presented for 2 s. After 1 s, either action 1 (magenta) or action 2 (green) is selected based on spiny projection neuron (SPN) activity. A correct trial is one in which action 1 (resp. 2) is selected after stimulus 1 (resp. 2) is presented. Second row: firing rates of four SPNs. Dark and light colors denote SPNs that represent actions 1 and 2, respectively. Third and fourth rows: projection of SPN activity onto difference and sum modes for actions 1 and 2. (**B**) Same as A, but illustrating the first trial, in which stimulus 2 is presented and action 1 is incorrectly selected. (**C**) Same as B, but illustrating the last trial, in which stimulus 1 is presented and action 1 is correctly selected.

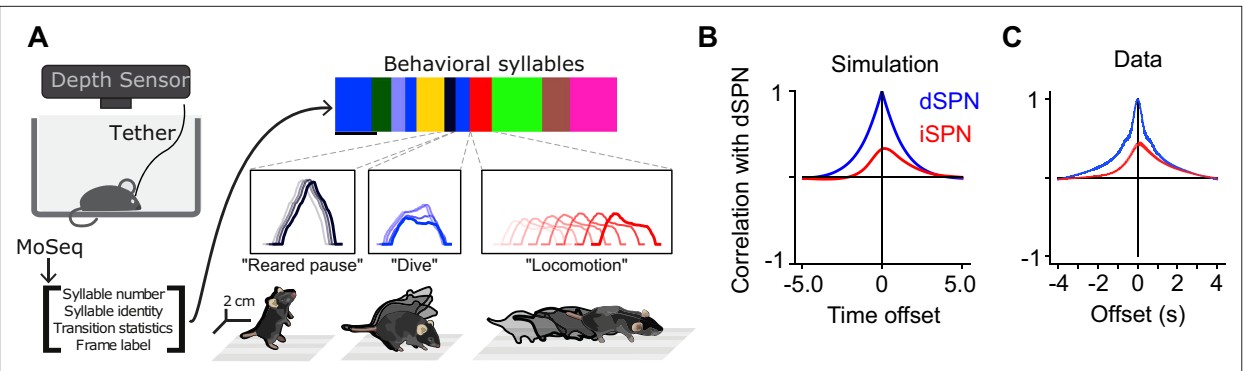

**Figure 5.** Comparisons of model predictions about bulk dSPN and iSPN activity to experimental data. (**A**) Schematic of experimental setup, taken from *Markowitz et al., 2018*. Neural activity and kinematics of spontaneously behaving mice are recorded, and behavior is segmented into stereotyped 'behavioral syllables' using the MoSeq pipeline. (**B**) In simulation of efference model with random feedforward cortical inputs, cross-correlation of total dSPN and iSPN activity. (**C**) Cross-correlation between fiber photometry recordings of bulk dSPN and iSPN activity in freely behaving mice, using the data from *Markowitz et al., 2018*. Line thickness indicates standard error of the mean.

The online version of this article includes the following figure supplement(s) for figure 5:

**Figure supplement 1.** Reversed indicator analysis.

populations of dSPNs and iSPNs promote or suppress different actions, (2) the feedforward inputs to all SPNs are random, (3) actions are sampled with log-likelihoods scaled according to the associated dSPN and iSPN difference mode, and (4) efferent activity excites the sum mode corresponding to the chosen action.

In this model, difference mode dSPN and iSPN activity drives behaviors, and those behaviors cause efferent activation of the corresponding sum mode. As a result, on average, dSPN activity tends to lead to increased future iSPN activity, while iSPN activity leads to decreased future dSPN activity. Consequently, the temporal cross-correlation between total dSPN and iSPN activity is asymmetric, with present dSPN activity correlating more strongly with future iSPN activity than with past iSPN activity (*Figure 5B*). Such asymmetry is not predicted by the canonical action selection model, or models that assume dSPNs and iSPNs are co-active. Computing the temporal cross-correlation in the bulk two-color photometry recordings of dSPN and iSPN activity, we find a very similar skewed relationship in the data (*Figure 5C*). We confirmed this result is not an artifact of the use of different indicators for dSPN and iSPN activity by repeating the analysis on data from mice where the indicators were reversed and finding the same result (*Figure 5—figure supplement 1*).

Our model makes even stronger predictions about SPN population activity and its relationship to action selection. First, it predicts that both dSPNs and iSPNS exhibit similar selectivities in their tuning to actions. This contrasts with implementations of the canonical action selection model in which iSPNs are active whenever their associated action is not being performed and thus are more broadly tuned than dSPNs (*Figure 1A*). Second, it also predicts that efferent activity excites dSPNs that promote the currently performed action and iSPNs that suppress the currently performed action. As a result, dSPNs whose activity increases during the performance of a given action should tend to be above baseline shortly prior to the performance of that action. By contrast, iSPNs whose activity increases during an action should tend to be below baseline during the same time interval (*Figure 6A*, left; *Figure 4C*). Moreover, this effect should be action-specific: the dSPNs and iSPNs whose activity increases during a given action should display negligible average fluctuations around the onset of other actions (*Figure 6A*, right). These predictions can also be reinterpreted in terms of the sum and difference modes. The difference mode activity associated with an action is elevated prior to selection of that action, while the sum mode activity is excited following action selection (*Figures 4C and 6B*). These two phases of difference and sum mode activity are not predicted by the canonical action selection model.

To test these hypotheses, we used calcium imaging data collected during spontaneous mouse behavior (*Markowitz et al., 2018*). The behavior of the mice was segmented into consistent, stereotyped kinematic motifs referred to as 'behavioral syllables', as in previous studies (*Figure 5A*). We regard these behavioral syllables as the analogs of actions in our model. First, we examined the tuning

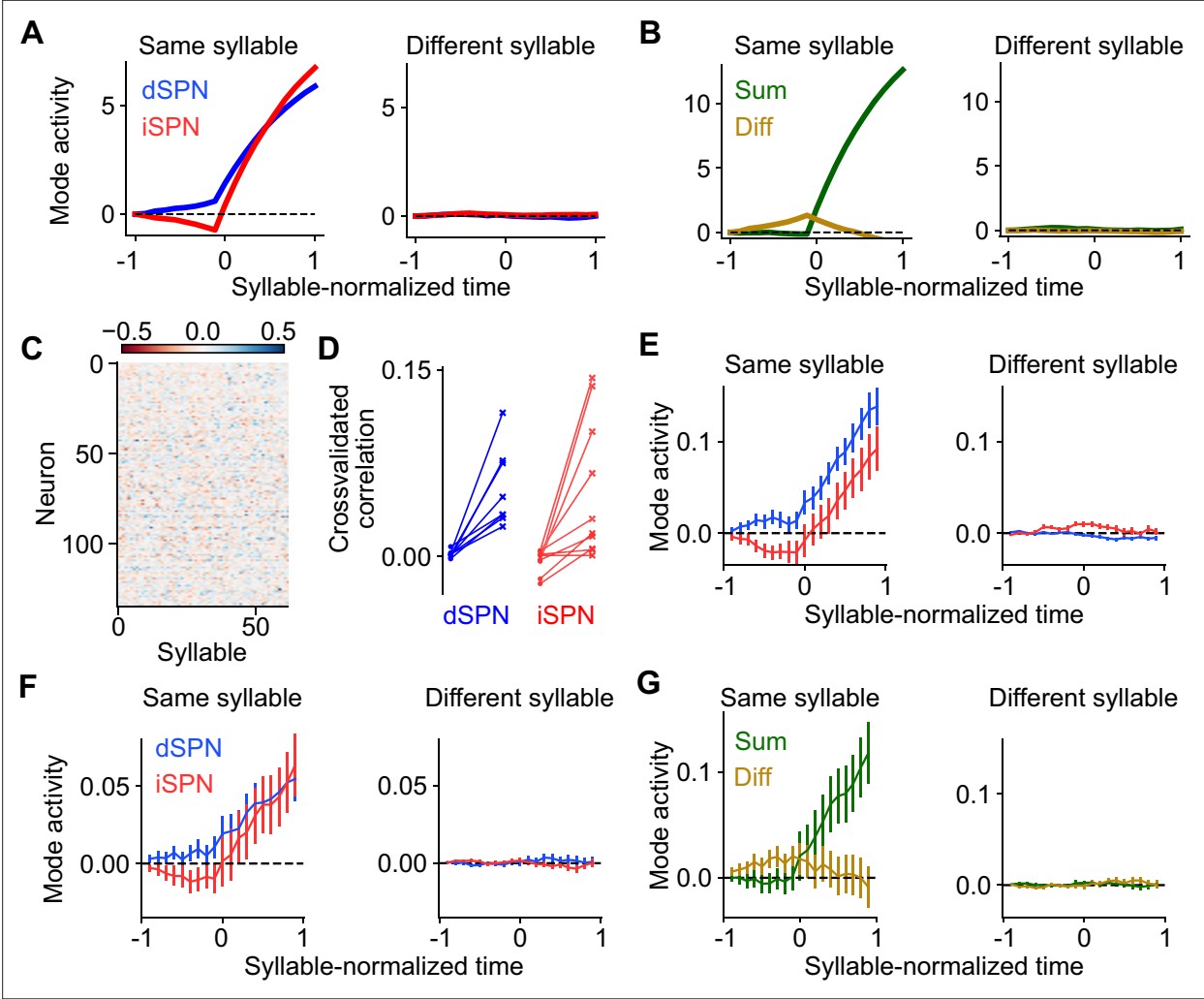

**Figure 6.** Comparisons of model predictions about action-tuned spiny projection neuron (SPN) subpopulations to experimental data. (**A**) Activity of dSPNs (blue) and iSPNs (red) around the onset of their associated action (left) or other actions (right) in the simulation from *Figure 5*. (**B**) Same information as A, but plotting activity of the sum (dSPN + iSPN) and difference (dSPN − iSPN) modes. (**C**) For an example experimental session, dSPN activity modes associated with each of the behavioral syllables, in *z*-scored firing rate units. (**D**) Correlation between identified dSPN and iSPN activity modes in two random subsamples of the data, for shuffled (left, circles) and real (right, x's) data. (**E**) Projection of dSPN (blue) and iSPN (red) activity onto the syllable-associated modes identified in panel C, around the onset of the associated syllable (left panel) or other syllables (right panel) averaged across all syllables. Error bars indicate standard error of the mean across syllables. (**F**) Same as panel E, restricting the analysis to mice in which dSPNs and iSPNs were simultaneously recorded. (**G**) Same data as panel F, but plotting activity of the sum (dSPN + iSPN) and difference (dSPN − iSPN) modes.

The online version of this article includes the following figure supplement(s) for figure 6:

**Figure supplement 1.** Comparison of dSPN and iSPN tuning selectivity.

of dSPNs and iSPNs to different actions and found that, broadly consistent with what our model predicts, both subpopulations exhibit similar selectivities (*Figure 6—figure supplement 1*). Next, to test our predictions about dynamics before and after action selection (*Figure 6A, B*), we identified, for each syllable, dSPN and iSPN population activity vectors (modes) that increased the most during performance of that syllable (*Figure 6C*). We confirmed that these modes are meaningful by checking that modes identified using two disjoint subsets of the data are correlated (*Figure 6D*). We then plotted the activity of these modes around the time of onset of the corresponding syllable, and averaged the result across the choice of syllables (*Figure 6E*). The result displays remarkable agreement with the model prediction in *Figure 6A*.

The majority of the above data consisted of recordings of either dSPNs or iSPNs from a given mouse. However, in a small subset (*n* = 4) of mice, dSPNs and iSPNs were simultaneously recorded

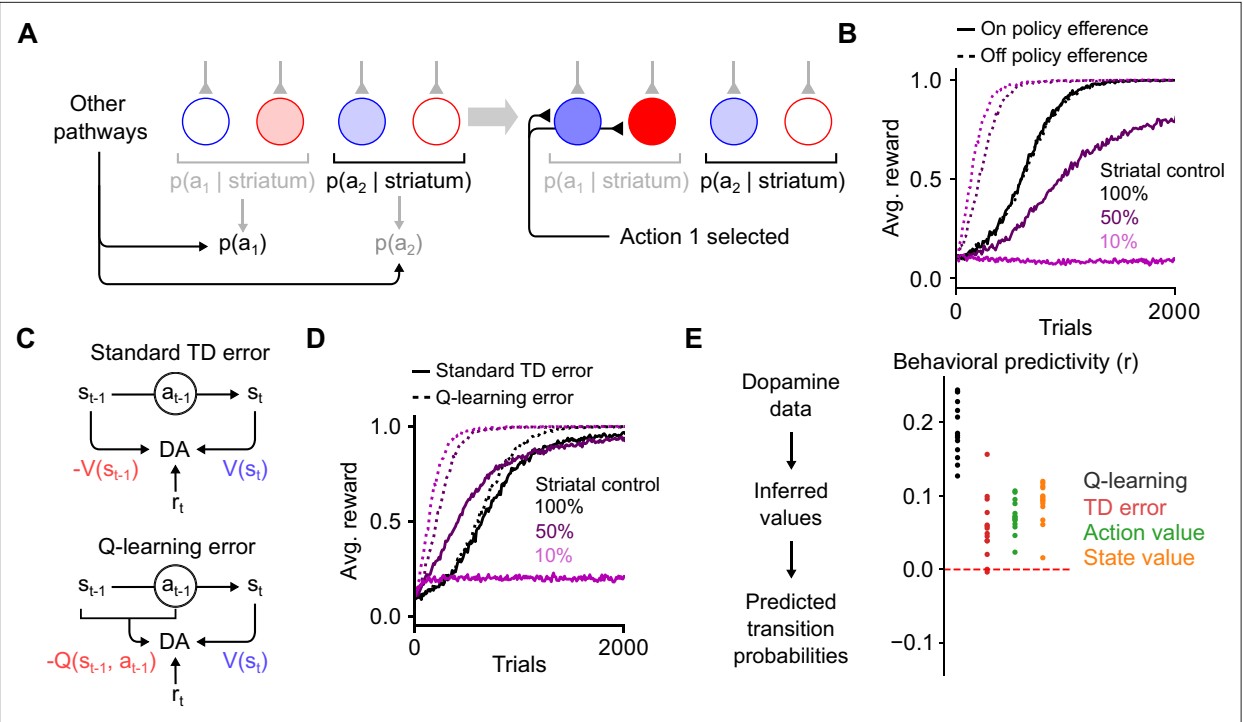

**Figure 7.** The efference model enables off-policy reinforcement learning. (**A**) Illustration of the efference model when the striatum shares control of behavior with other pathways. In this example, striatal activity biases the action selection toward choosing action 2, but other neural pathways override the striatum and cause action 1 to be selected instead (left). Following action selection, efferent activity excites the dSPN and iSPN associated with action 1. However, the action probability readouts of the striatal population remain unchanged. (**B**) Performance of RL models in a simulated action selection task (10 cortical states, 10 available actions, in each state one of the actions results in a reward of 1 and the others result in zero reward). Control is shared between the striatal RL circuit and another pathway that biases action selection toward the correct action. Different lines indicate different strength of striatal control relative to the strength of the other pathway. Line style (dashed or solid) indicates the efference model: off-policy efference excites spiny projection neurons (SPNs) associated with the selected action, while on-policy efference excites SPNs associated with the action most favored by the striatum. (**C**) Schematic of different reinforcement learning models of dopamine activity. The standard temporal difference (TD) error model predicts that dopamine activity is sensitive to reward, the predicted value of the current state, and the predicted value of the previous state. The Q-learning error model predicts sensitivity to reward, the predicted value of the current state, and the predicted value of the previous state–action pair. (**D**) In the task of panel B using the off-policy efference model, comparison between different models of dopamine activity as striatal control is varied (the Q-learning error model was used in panel B). (**E**) Correlation between predicted and actual syllable-to-syllable transition matrix. Predictions were made according to different models of the relationship between dopamine activity and behavior, using observed average dopamine activity associated with syllable transitions in the data of *Markowitz et al., 2023*. Each dot indicates a different experimental session.

The online version of this article includes the following figure supplement(s) for figure 7:

**Figure supplement 1.** Comparison to counterfactual model in which iSPNs use the same plasticity rule as dSPNs.

and identified. We repeated the analysis above on these sessions, and found the same qualitative results (*Figure 6F*). The simultaneous recordings further allowed us to visualize the sum and difference mode activity (*Figure 6G*), which also agrees with the predictions of our model (*Figure 6B*).

## Efference model enables off-policy reinforcement learning

Prior studies have argued for the importance of motor efference copies during basal ganglia learning, in particular when action selection is influenced by other brain regions (*Fee, 2014*; *Lindsey and Litwin-Kumar, 2022*). Indeed, areas such as the motor cortex and cerebellum drive behavior independent of the basal ganglia (*Exner et al., 2002*; *Wildgruber et al., 2001*; *Ashby et al., 2010*; *Silveri, 2021*; *Bostan and Strick, 2018*). Actions taken by an animal may therefore at times differ from those most likely to be selected by striatal outputs (*Figure 7A*), and it may be desirable for corticostriatal synapses to learn about the consequences of these actions.

In the reinforcement learning literature, this kind of learning is known as an 'off-policy' algorithm, as the reinforcement learning system (in our model, the striatum) learns from actions that follow a

different policy than its own. Off-policy learning has been observed experimentally, for instance in the consolidation of cortically driven behaviors into subcortical circuits including DLS (*Kawai et al., 2015*; *Hwang et al., 2019*; *Mizes et al., 2023*). Such learning requires efferent activity in SPNs that reflects the actions being performed, rather than the action that would be performed based on the striatum's influence alone.

We modeled this scenario by assuming that action selection is driven by weighted contributions from both the striatum and other motor pathways and that the ultimately selected action drives efferent activity (*Figure 7A*; see Methods). We found that when action selection is not fully determined by the striatum, such efferent activity is critical for successful learning (*Figure 7B*). Notably, in our model, efferent activity has no effect on striatal action selection, due to the orthogonality of the sum and difference modes (*Figure 3B*). In a hypothetical alternative model in which the iSPN plasticity rule is the same as that of dSPNs, the efferent activity needed for learning is not orthogonal to the output of the striatum, impairing off-policy learning (*Figure 7—figure supplement 1*). Thus, efferent excitation of opponent dSPNs/iSPNs is necessary both to implement correct learning updates given dSPN and iSPN plasticity rules, and to enable off-policy reinforcement learning.

## Off-policy reinforcement learning predicts relationship between dopamine activity and behavior

We next asked whether other properties of striatal dynamics are consistent with off-policy reinforcement learning. We focused on the dynamics of dopamine release, as off-policy learning makes specific predictions about this signal. Standard TD learning models of dopamine activity (*Figure 7C*, top) determine the expected future reward (or 'value') $V(s)$ associated with each state $s$ using the following algorithm:

$$\delta_t = r_t + V(s_t) - V(s_{t-1}), \tag{4}$$

$$V(s_t) \leftarrow V(s_t) + \alpha \delta_t, \tag{5}$$

where $s_t$ and $s_{t-1}$ indicate current and previous states, $r_t$ indicates the currently received reward, $\alpha$ is a learning rate factor, and $\delta_t$ is the TD error thought to be reflected in phasic dopamine responses. These dopaminergic responses can be used as the learning signal for a updating action selection in dorsal striatum (*Equations 1 and 2*), an arrangement commonly referred to as an 'actor-critic' architecture (*Niv, 2009*).

TD learning of a value function $V(s)$ is an on-policy algorithm, in that the value associated with each state is calculated under the assumption that the system's future actions will be similar to those taken during learning. Hence, such algorithms are is poorly suited to training an action selection policy in the striatum in situations where the striatum does not fully control behavior, as the values $V(s)$ will not reflect the expected future reward associated with a state if the striatum were to dictate behavior on its own. Off-policy algorithms such as Q-learning solve this issue by learning an action-dependent value function $Q(s, a)$, which indicates the expected reward associated with taking action $a$ in action $s$ (*Figure 7C*, bottom), via the following algorithm:

$$\delta_t = r_t + V(s_t) - Q(s_{t-1}, a_{t-1}), \tag{6}$$

$$V(s) = \max_a Q(s, a). \tag{7}$$

This algorithm predicts that the dopamine response $\delta_t$ is action-dependent. The significance of on- versus off-policy learning algorithms can be demonstrated in simulations of operant conditioning tasks in which control of action selection is shared between the striatum and another 'tutor' pathway that biases responses toward the correct action. When the striatal contribution to decision-making is weak, it is unable to learn the appropriate response when dopamine activity is modeled as a TD error (*Figure 7D*). On the other hand, a Q-learning model of dopamine activity enables efficient striatal learning even when control is shared with another pathway.

For the spontaneous behavior paradigm we analyzed previously (*Figure 5A*), Q-learning but not TD learning of $V(s)$ predicts sensitivity of dopamine responses to the likelihood of the previous syllable-to-syllable transition. Using recordings of dopamine activity in the DLS in this paradigm (*Markowitz et al., 2023*), we tested whether a Q-learning model could predict the relationship between dopamine activity and behavioral statistics, comparing it to TD learning of $V(s)$ and other alternatives (see

Appendix). The Q-learning model matches the data significantly better than alternatives (*Figure 7E*), providing support for a model of dorsal striatum as an off-policy reinforcement learning system.

## Discussion

We have presented a model of reinforcement learning in the dorsal striatum in which efferent activity excites dSPNs and iSPNs that promote and suppress, respectively, the currently selected action. Thus, following action selection, iSPN activity counterintuitively represents the action that is inhibited by the currently active iSPN population. This behavior contrasts with previous proposals in which iSPN activity reflects actions being inhibited. This model produces updates to corticostriatal synaptic weights given the known opposite-sign plasticity rules in dSPNs and iSPNs that correctly implement a form of reinforcement learning (*Figure 3*), which in the absence of such efferent activity produce incorrect weight updates (*Figure 2*). The model makes several novel predictions about SPN activity which we confirmed in experimental data (*Figures 5 and 6*). It also enables multiplexing of action selection signals and learning signals without interference. This facilitates more sophisticated learning algorithms such as off-policy reinforcement learning, which allows the striatum to learn from actions that were driven by other neural circuits. Off-policy reinforcement learning requires dopamine to signal action-sensitive reward predictions errors, which agrees better with experimental recordings of striatal dopamine activity than alternative models (*Figure 7*).

### Other models of striatal action selection

Prior models have modeled the opponent effects of dopamine on dSPN and iSPN plasticity (*Frank, 2005*; *Collins and Frank, 2014*; *Jaskir and Frank, 2023*). In these models, dSPNs come to represent the positive outcomes and iSPNs the negative outcomes associated with a stimulus–action pair. Such models can also represent uncertainty in reward estimates (*Mikhael and Bogacz, 2016*). Appropriate credit assignment in these models requires that only corticostriatal weights associated with SPNs encoding the chosen action are updated. Our model clarifies how the neural activity required for such selective weight updates can be multiplexed with the neural activity required for action selection, without requiring separate phases for action selection and learning.

*Bariselli et al., 2019* also argue against the canonical action selection model and propose a competitive role for dSPNs and iSPNs that is consistent with our model. However, the role of efferent activity and distinctions between action- and learning-related signals are not discussed.

Our model is related to these prior proposals but identifies motor efference as key for appropriate credit assignment across corticostriatal synapses. It also provides predictions concerning the temporal dynamics of such signals (*Figure 4*) and a verification of these using physiological data (*Figure 7*).

### Other models of efferent inputs to the striatum

Prior work has pointed out the need for efference copies of decisions to be represented in the striatum, particularly for actions driven by other circuits (*Fee, 2014*). *Frank, 2005* proposes a model in which premotor cortex outputs collateral signals to the striatum that represent the actions under consideration, with the striatum potentially biasing the decision based on prior learning. Through bidirectional feedback (premotor cortex projecting to striatum, and striatum projecting to premotor cortex indirectly through the thalamus) a decision is collectively made by the combined circuit, and the selected action is represented in striatal activity, facilitating learning about the outcome of the action. While similar to our proposal in some ways, this model implicitly assumes that the striatal activity necessary for decision-making is also what is needed to facilitate learning. As we point out in this work, due to the opponent plasticity rules in dSPNs and iSPNs, a post hoc efferent signal that is not causally relevant to the decision-making process is necessary for appropriate learning.

Other authors have proposed models in which efferent activity is used for learning. In the context of vocal learning in songbirds, *Fee and Goldberg, 2011* proposed that the variability-generating area LMAN, which projects to the song motor pathway, sends collateral projections to Area X, which undergoes dopamine-modulated plasticity. In this model, the efferent inputs to Area X allow it to learn which motor commands are associated with better song performance (signaled by dopamine). Similar to our model, this architecture implements off-policy reinforcement learning in Area X, with HVC inputs to Area X being analogous to corticostriatal projections in our model. However,

in our work, the difference in plasticity rules between dSPNs and iSPNs is key to avoiding interference between efferent learning-related activity and feedforward action selection-related activity. A similar architecture was proposed in *Fee, 2012* in the context of oculomotor learning, in which oculomotor striatum receives efferent collaterals from the superior colliculus and/or cortical areas which generate exploratory variability. *Lisman, 2014* also propose a high-level model of striatal efferent inputs similar to ours, and also point out the issue with the iSPN plasticity rule assigning credit to inappropriate actions without efferent inputs. *Rubin et al., 2021* argue that sustained efferent input is necessary for temporal credit assignment when reward is delayed relative to action selection.

Our model is consistent with these prior proposals, but describes how efferent input must be targeted to opponent SPNs. In our work, the distinction between dSPN and iSPN plasticity rules is key to enable multiplexing of action selection and efferent learning signals without interference. Previous authors have proposed other mechanisms to avoid interference. For instance, *Fee, 2014* proposes that efferent inputs might influence plasticity without driving SPN spiking by synapsing preferentially onto dendritic shafts rather than spines. To avoid action selection-related spikes interfering with learning, the system may employ spike-timing-dependent plasticity rules that are tuned to match the latency at which efferent inputs excite SPNs. While these hypotheses are not mutually exclusive to ours, our model requires no additional circuitry or assumptions beyond the presence of appropriately tuned efferent input (see below) and opposite-sign plasticity rules in dSPNs and iSPNs, due to the orthogonality of the sum and difference modes. An important capability enabled by our model is that action selection and efferent inputs can be multiplexed simultaneously, unlike the works cited above, which posit the existence of temporally segregated action selection and learning phases of SPN activity.

## Biological substrates of striatal efferent inputs

Efferent inputs to the striatum must satisfy two important conditions for our model to learn correctly. Neither of these has been conclusively demonstrated, and the two conditions thus represent predictions or assumptions necessary for our model to function. First, they must be appropriately targeted: when an action is performed, dSPNs and iSPNs associated with that action must be excited, but other dSPNs and iSPNs must not be. The striatum receives topographically organized inputs from cortex (*Peters et al., 2021*) and thalamus (*Smith et al., 2004*), with neurons in some thalamic nuclei exhibiting long-latency responses (*Minamimoto et al., 2005*). SPNs tuned to the same behavior tend to be located nearby in space (*Barbera et al., 2016*; *Shin et al., 2020*; *Klaus et al., 2017*). This anatomical organization could enable action-specific efferent inputs. We note that this does not require a spatially specific dopaminergic signal (*Wärnberg and Kumar, 2023*). In our models, we assume that dopamine conveys a global, scalar prediction error. Another possibility is that targeting of efferent inputs could be tuned via plasticity during development. For instance, if a dSPN promotes a particular action, reward-independent Hebbian plasticity of its efferent inputs would potentiate those inputs that encode the promoted action. Reward-independent anti-Hebbian plasticity would serve an analogous function for iSPNs. Alternatively, if efferent inputs are fixed, plasticity downstream of striatum could adapt the causal effect of SPNs to match their corresponding efferent input.

A second key requirement of our model is that efferent input synapses should not be adjusted according to the same reward-modulated plasticity rules as the feedforward corticostriatal inputs, as these rules would disrupt the targeting of efferent inputs to the corresponding SPNs. This may be achieved in multiple ways. One possibility is that efferent inputs project from different subregions or cell types than feedforward inputs and are subject to different forms of plasticity. Alternatively, efferent input synapses may have been sufficiently reinforced that they exist in a less labile, 'consolidated' synaptic state. A third possibility is that the system may take advantage of latency in efferent activity. Spike timing dependence in SPN input plasticity has been observed in several studies (*Shen et al., 2008*; *Fino et al., 2005*; *Pawlak and Kerr, 2008*; *Fisher et al., 2017*). This timing dependence could make plasticity sensitive to paired activity in state inputs and SPNs while being insensitive to paired activity in efferent inputs and SPNs. Investigating the source of efferent inputs to SPNs and how it is differentiated from other inputs is an important direction for future work.

## Extensions and future work

We have assumed that the striatum selects among a finite set of actions, each of which corresponds to mutually uncorrelated patterns of SPN activity. In reality, there is evidence that the striatal code for action is organized such that kinematically similar behaviors are encoded by similar SPN activity patterns (*Klaus et al., 2017*; *Markowitz et al., 2018*). Other work has shown that the DLS can exert influence over detailed kinematics of learned motor behaviors, rather than simply select among categorically distinct actions (*Dhawale et al., 2021*). A more continuous, structured code for action in DLS is useful in allowing reinforcement learning to generalize between related actions. The ability afforded by our model to multiplex arbitrary action selection and learning signals may facilitate these more sophisticated coding schemes. For instance, reinforcement learning in continuous-valued action spaces requires a three-factor learning rule in which the post-synaptic activity factor represents the discrepancy between the selected action and the action typically selected in the current behavioral state (*Lindsey and Litwin-Kumar, 2022*), which in our model would be represented by efferent activity in SPNs. Investigating such extensions to our model and their consequences for SPN tuning is an interesting future direction.

In this work, we find strong empirical evidence for our model of efferent activity in SPNs and show that in principle it enables off-policy reinforcement learning capabilities. A convincing experimental demonstration of off-policy learning capabilities would require a way of identifying the causal contribution of SPN activity to action selection, in order to distinguish between actions that are consistent (on-policy) or inconsistent (off-policy) with SPN outputs. This could be achieved through targeted stimulation of SPN populations, or by recording SPN activity during behaviors that are known to be independent of striatal influence (*Mizes et al., 2023*). Simultaneous recordings in SPNs and other brain regions would also facilitate distinguishing between actions driven by striatum from those driven by other pathways. Our model predicts that the relative strength of fluctuations in difference mode versus sum mode activity should be greatest during striatum-driven actions. Such experimental design would also enable a stronger test of the Q-learning model of dopamine activity: actions driven by other regions should lead to increased dopamine activity, as they will be predicted according to the striatum's learned action values to have low value.

In our model, the difference between dSPN and iSPN plasticity rules is key to enabling multiplexing of action selection and learning-related activity without interference. Observed plasticity rules elsewhere in the brain are also heterogeneous; for instance, both Hebbian and anti-Hebbian weight changes are observed in cortico-cortical connections (*Koch et al., 2013*; *Chindemi et al., 2022*). It is an interesting question whether a similar strategy may be employed outside the striatum, and in other contexts besides reinforcement learning, to allow simultaneous encoding of behavior and learning-related signals without interference.

## Methods

### Numerical simulations

Code implementing the model is available on GitHub (https://github.com/alitwinkumar/lindsey_etal_striatal_dynamics, copy archived at *Litwin-Kumar, 2025*).

### Data availability

We reanalyzed data from *Markowitz et al., 2018* and *Markowitz et al., 2023*. Data from *Markowitz et al., 2023* is available at https://dx.doi.org/10.5281/zenodo.7274802.

### Basic model architecture

In our simulated learning tasks, we used networks with the following architecture. SPNs receive inputs from cortical neurons. In our simulated go/no-go tasks, there is a single cortical input neuron (representing a task cue) with activity equal to 1 on each trial. In simulated tasks with multiple different task cues (such as the two-alternative forced choice task), there is a population of cortical input neurons, each of which is active with activity 1 when the corresponding task cue is presented and 0 otherwise. The task cue is randomly chosen with uniform probability each trial.

For each of the $A$ actions available to the model, there is an assigned dSPN and iSPN. We choose to use a single neuron per action for simplicity of the model, but our model could easily be generalized

to use population activity to encode actions. The activities of the dSPN and iSPN associated with action $a$ are denoted as $y_a^{\text{dSPN}}$ and $y_a^{\text{iSPN}}$, respectively. Each dSPN and iSPN receives inputs from $M$ cortical neurons, and the synaptic input weights from cortical neuron $j$ to the dSPN or iSPN associated with action $a$ are denoted as $w_{aj}^{\text{dSPN}}$ or $w_{aj}^{\text{iSPN}}$. Feedforward SPN activity is given by

$$y_a^{\text{dSPN}} = \phi \left( \sum_{j=1}^{M} w_{aj}^{\text{dSPN}} x_j \right), \tag{8}$$

$$y_a^{\text{iSPN}} = \phi \left( \sum_{j=1}^{M} w_{aj}^{\text{iSPN}} x_j \right), \tag{9}$$

where $\phi$ is a nonlinear activation function. We choose $\phi$ to be the rectified linear function: $\phi(h) = \max(0, h)$.

Action selection depends on SPN activity in the following manner. The log-likelihood of an action $a$ being performed is proportional to $\ell_a = y_a^{\text{dSPN}} - y_a^{\text{iSPN}}$. That is, dSPN activity increases the likelihood of taking the action and iSPN activity decreases the likelihood of taking the action. Concretely, the probability of action $a$ being taken is:

$$p(a) = \frac{e^{\beta \ell_a}}{c_{\text{no-go}} + \sum_{a'} e^{\beta \ell_{a'}}}, \tag{10}$$

where $\beta$ is a parameter controlling the degree of stochasticity in action selection (higher $\beta$ corresponds to more deterministic choices), and $c_{\text{no-go}}$ controls the probability that no action is taken. In the simulated go/no-go tasks we choose $c_{\text{no-go}} = 1$ and in the tasks involving selection among multiple actions we choose $c_{\text{no-go}} = 0$. Except where otherwise noted we used $\beta = 10.0$ in all task simulations.

## Models of SPN activity following action selection

In the 'canonical action selection model' (**Figure 1**), following action selection, the activity of the dSPN associated with the selected action and the activity of all iSPNs associated with unselected actions are set to 1. Biologically, this activity pattern can be implemented via effective mutual inhibition between SPNs with opponent functions (dSPNs tuned to different actions, iSPNs tuned to different actions, and dSPN/iSPN pairs tuned to the same action) and mutual excitation between SPNs with complementary functions (dSPNs tuned to one action and iSPNs to another) (**Burke et al., 2017**).

In the proposed efference model, following selection of an action $a^*$, activity of the SPNs associated with action $a^*$ is updated as follows:

$$y_a^{\text{dSPN}} \leftarrow \phi \left( c_{\text{efference}} \cdot 1[a = a^*] + \sum_{j=1}^{M} w_{aj}^{\text{dSPN}} x_j \right), \tag{11}$$

$$y_a^{\text{iSPN}} \leftarrow \phi \left( c_{\text{efference}} \cdot 1[a = a^*] + \sum_{j=1}^{M} w_{aj}^{\text{iSPN}} x_j \right), \tag{12}$$

where $1[a = a^*]$ equals 1 for $a = a^*$ and 0 otherwise. The parameter $c_{\text{efference}}$ controls the strength of efferent excitation.

## Learning rules

In all models, SPN input weights are initialized at 1 and weight updates proceed according to the plasticity rules given below:

$$\Delta w_{aj}^{\text{dSPN}} = \alpha \left( f^{\text{dSPN}}(\delta) \cdot y_a^{\text{dSPN}} \cdot x_j \right), \tag{13}$$

$$\Delta w_{aj}^{\text{iSPN}} = \alpha \left( f^{\text{iSPN}}(\delta) \cdot y_a^{\text{iSPN}} \cdot x_j \right), \tag{14}$$

where $\alpha$ is a learning rate, set to 0.05 throughout all learning simulations except the tutoring simulations of *Figure 7* where it is set to 0.01. In the paper we experiment with various choices of $f^{\mathrm{dSPN}}$ and $f^{\mathrm{iSPN}}$.

$$f^{\mathrm{dSPN}}(\delta) = \delta, \, f^{\mathrm{iSPN}}(\delta) = -\delta \quad \text{(Linear)}, \tag{15}$$

$$f^{\mathrm{dSPN}}(\delta) = \max(\delta, 0), \, f^{\mathrm{iSPN}}(\delta) = \max(-\delta, 0) \quad \text{(Rectified)}, \tag{16}$$

$$f^{\mathrm{dSPN}}(\delta) = \frac{1}{2}\left(a + \left(\frac{b}{1 + ce^{1-d\delta}}\right)\right), \, f^{\mathrm{iSPN}}(\delta) = \frac{1}{2}\left(a + \left(\frac{b}{1 + ce^{1+d\delta}}\right)\right) \text{ (Offset sigmoid)}, \tag{17}$$

with the offset sigmoid parameters chosen as $a = -3.5, b = 11.5, c = 0.9, d = 1$ (taken from *Cruz et al., 2022*). The quantity $\delta$ indicates an estimate of reward prediction error. In our experiments in *Figures 2 and 3* we use TD learing to compute $\delta$:

$$\delta = r - V(s), \tag{18}$$

$$\Delta V(s) = \alpha_V \delta, \tag{19}$$

where $\alpha_V$ is a learning rate, set to 0.05 throughout all learning simulations (except the tutoring simulations of *Figure 7* where it is set to 0.25) and $s$ indicates the cortical input state (indicating which cue is being presented). $V(s)$ is initialized at 0.

In our experiments in *Figure 7* we use Q-learning to enable off-policy learning, corresponding to the following value for $\delta$:

$$\delta = r - Q(s, a), \tag{20}$$

where $a$ indicates the action that was just taken in response to state $s$, and $Q(s, a)$ is taken to be equal to the striatal output $\ell_a = y_a^{\mathrm{dSPN}} - y_a^{\mathrm{iSPN}}$ in response to the state $s$.

## Firing rate simulations

In each trial of the two-alternative forced choice task (*Figure 4*), one of two stimuli is presented for 2 s. Cortical activity $\mathbf{x}$ representing the stimulus is encoded in a one-hot vector. Four SPNs are modeled, one dSPN and one iSPN for each of two actions. The dynamics of SPN $i$ follows:

$$\tau \frac{dy_i}{dt} = -y_i + \left[\sum_j w_{ij}x_j + \eta_i(t) + e_i(t) + b\right]_+. \tag{21}$$

Here, $[\cdot]_+$ denotes positive rectification, $w_{ij}$ represent corticostriatal weights initialized following a Gaussian distribution with mean 0 and standard deviation 1 Hz, $\eta_i(t)$ is an Ornstein–Uhlenbeck noise process with time constant 600 ms and variance 1/60 Hz$^2$, $e_i(t)$ denotes efferent input, and $b = 5$ Hz is a bias term. Simulations were performed with $dt = 20$ ms.

On each trial, an action is selected based on the average difference mode activity for the two actions during the first 1 s of stimulus presentation. In the second half of the stimulus presentation period, efferent input is provided to the dSPN and iSPN corresponding to the chosen action by setting $e_i(t) = 7.5$ Hz for these neurons. Learning proceeds according to

$$\frac{dw_{ij}}{dt} = \eta f_i(\delta)(y_i(t) - b)x_j(t), \tag{22}$$

where in the second half of the stimulus presentation period $f_i(\delta) = 1$ for dSPNs after a correct action is taken and iSPNs after an incorrect action is taken, and –1 otherwise, and $\eta = 5 \times 10^{-4}$ ms$^{-1}$.

## Experimental prediction simulations

For the model predictions of *Figures 5 and 6*, we used the following parameters: $A = 50$, $\beta = 100$, $c_{\mathrm{efference}} = 1.5$, and we set $c_{\mathrm{no-go}}$ such that the no-action option was chosen 50% of the time. Feedforward SPN activity was generated from a Gaussian process with kernel $k(t_1, t_2) = e^{-|t_1-t_2|/10}$ (exponentially decaying autocorrelation with a time constant of 10 timesteps). Efference activity also decayed exponentially with a time constant of 10 timesteps. Action selection occured every 10 timesteps based on the SPN activity at the preceding timestep.

## Fiber photometry data

Adeno-associated viruses (AAVs) expressing Cre-On jRCaMP1b and Cre-Off GCaMP6s were injected into the DLS of $n = 10$ *Drd1a-Cre* mice to measure bulk dSPN (red) and iSPN (green) activity via multicolor photometry. Activity of each indicator was recorded at a rate of 30 Hz using an optical fiber implanted in the right DLS. Data was collected during spontaneous behavior in a circular open field, for five to six sessions of 20 min each for each mouse. In the reversed indicator experiments of *Figure 5—figure supplement 1*. *A2a-Cre* mice were injected with a mixture of the same AAVs, labeling iSPNs with jRCaMP1b (red) and dSPNs with GCaMP6s (green). More details are reported in *Markowitz et al., 2018*.

In our data analyses in *Figure 5C*, *Figure 5—figure supplement 1*, for each session ($n = 48$ and $n = 8$, respectively) we computed the autocorrelation and cross-correlation of the dSPN and iSPN indicator activity across the entire session.

## Miniscope data

*Drd1a-Cre* AAVs expressing GCaMP6f were injected into the right DLS of $n = 4$ *Drd1a-Cre* mice (to label dSPNs) and $n = 6$ *A2a-Cre* mice (to label iSPNs). A head-mounted single-photon microscope was coupled to a gradient index lens implanted into the dorsal striatum above the injection site. Recordings were made, as for the photometry data, during spontaneous behavior in a circular open field. Calcium activity was recorded from a total of 653 dSPNs and 794 iSPNs for these mice, with the number of neurons per mouse ranging from 27 to 336. To enable simultaneous recording of dSPNs and iSPNs in the same mice, a different protocol was used: *Drd1a-Cre* mice were injected with an AAV mixture which labeled both dSPNs and iSPNS with GCaMP6s, but additionally selectively labeled dSPNS with nuclear-localized dTomato. This procedure enabled (in $n = 4$ mice) cell-type identification of dSPNs versus iSPNs with a two-photon microscope which was cross-referenced with the single-photon microscope recordings. More details are given in *Markowitz et al., 2018*. In our analyses, these data were used for the simultaneous-recording analyses in *Figure 6F,G* and were also combined with the appropriate single-pathway data in the analyses of *Figure 6D,E*.

## Behavioral data

Mouse behavior in the circular open field was recorded as follows: 3D pose information was recorded using a depth camera at a rate of 30 Hz. The videos were preprocessed to center the mouse and align the nose-to-tail axis across frames and remove occluding objects. The videos were then fed through PCA to reduce the dimensinoality of the data and fed into the MoSeq algorithm (*Wiltschko et al., 2015*) which fits a generative model to the video data that automatically infers a set of behavioral 'syllables' (repeated, stereotyped behavioral kinematics) and assigns each frame of the video to one of these syllables. More details on MoSeq are given in *Wiltschko et al., 2015* and more details on its application to this dataset are given in *Markowitz et al., 2018*. There were 89 syllables identified by MoSeq that appear across all the sessions. We restricted our analysis to the set of 62 syllables that appear at least 5 times in each behavioral session.

## Syllable-tuned SPN activity mode analysis

In our analysis, we first *z*-scored the activity of each neuron across the data collected for each mouse. We divided the data by the boundaries of behavioral syllables and split it into two equally sized halves (based on whether the timestamp, rounded to the nearest second, of the behavioral syllable was even or odd). To compute the activity modes associated with each behavioral syllable, we first computed the average change in activity for each neuron during each syllable and fit a linear regression model to predict this increase from a one-hot vector indicating the syllable identity. The resulting coefficients of this regression indicate the directions (modes) in activity space that increase the most during performance of each of the behavioral syllables. We linearly time-warped the data in each session based on the boundaries of each MoSeq-identified behavioral syllable, such that in the new time coordinates each behavioral syllable lasted 10 timesteps. The time course of the projection of SPN activity along the modes associated with each behavioral syllable was then computed around the onset of that syllable, or around all other sllables. As a way of cross-validating the analysis, we performed the regression on one half of the data and plotted the average mode activity on the other half of the data (in both directions, and averaged the results). We averaged the resulting time courses of mode activity

across all choices of behavioral syllables. This analysis was performed for each mouse and the results in *Figure 6* show means and standard errors across mice.

## Comparison of selectivity of dSPNs and iSPNs

To test whether dSPNs or iSPNs exhibit greater or less specificity in their tuning to behaviors, we computed the selectivity of each neuron in the imaging data (*Figure 6—figure supplement 1*). For each neuron, we computed its average *z*-scored activity $a_i$ in response to each of the behavioral syllables $i \in \{1, ..., A\}$ in the dataset. Common measures of selectivity require a nonnegative measurement of a neuron's tuning to a given condition. Thus, we conducted the analysis in two ways, using either the unsigned activity $|a_i|$ or the rectified activity $\max(a_i, 0)$ as the measure of the neuron's tuning $t_i$ to syllable $i$. The selectivity was then computed using the following expression introduced in prior work (*Treves and Rolls, 1991*; *Willmore and Tolhurst, 2001*):

$$\frac{\left(\frac{1}{A}\sum_i t_i\right)^2}{\frac{1}{A}\sum_i t_i^2}. \tag{23}$$

This value ranges from 0 to 1, and a higher value indicates that fluctuations in a neuron's activity are driven primarily by one or a few behavioral syllables. The results are shown in *Figure 6—figure supplement 1*. The selectivity values are fairly modest (consistent with a distributed code for actions) and comparable between dSPNs and iSPNs.

## Dopamine activity data and analysis

For *Figure 7E*, we used data from *Markowitz et al., 2023*. Mice ($n = 14$) virally expressing the dopamine reporter dLight1.1 in the DLS were recorded with a fiber cannula implanted above the injection site. Mice were placed in a circular open field for 30-min sessions and allowed to behave freely while spontaneous dLight activity was recorded. MoSeq (described above) was used to infer a set of $S = 57$ behavioral syllables observed across all sessions. As in *Markowitz et al., 2023*, the data were preprocessed by computing the maximum dLight value during each behavioral syllable. These per-syllable dopamine values were *z*-scored across each session and used as our measure of dopamine activity during each syllable. We then computed an $S \times S$ table of the average dopamine activity during each syllable $s_t$ conditioned on the previous syllable having been syllable $s_{t-1}$, denoted as $D(s_{t-1}, s_t)$. We also computed the $S \times S$ table of probabilities of transitioning from syllable $s_{t-1}$ to syllable $s_t$ across the dataset, denoted as $P(s_{t-1}, s_t)$. These tables were computed separately for each mouse. In *Figure 7E*, we report the Pearson correlation coefficient between the predicted and actual values of $P(s_{t-1}, s_t)$. We then experimented with several alternative models, described below, that predict $P(s_{t-1}, s_t)$ based on $D(s_{t-1}, s_t)$. In *Figure 7E*, we report the Pearson correlation coefficient between the predicted and actual values of $P(s_{t-1}, s_t)$.

### Q-learning model

In the Q-learning model, the mouse maintains an internal estimate of the value $Q(s_{t-1}, s_t)$ of each transition between syllables. In the absence of explicit rewards, the dopamine activity associated with a syllable transition is predicted to be $D(s_{t-1}, s_t) = \max_{s'} Q(s_t, s') - Q(s_{t-1}, s_t)$. We inferred a set of $Q$-values by initializing a Q-table with all zero values and running gradient descent on the Q-table to minimize the mean squared error between the predicted and empirical values of $D(s_{t-1}, s_t)$. These inferred $Q$-values were used to predict behavioral transition probabilities according to $\hat{P}(s_{t-1}, s_t) = \frac{e^{\beta(s_{t-1})Q(s_{t-1}, s_t)}}{\sum_{s'} e^{\beta(s_{t-1})Q(s_{t-1}, s')}}$. We did not fit the value of $\beta(s_{t-1})$ but rather chose it to be the reciprocal of the standard deviation of $Q(s_{t-1}, s')$ across all $s'$, to ensure a reasonable dynamic range in predicted transition probabilities.

### *V(s)* TD learning model

In this model, the mouse maintains an internal estimate of the value $V(s)$ of each syllable, and the predicted dopamine activity at each transition is $D(s_{t-1}, s_t) = V(s_t) - V(s_{t-1})$. We fit the vector of values $V(s)$ to minimize the mean squared error of predicted and empirical $D(s_{t-1}, s_t)$. The predicted transition probabilities in this model (which are independent of the previous syllable $s_{t-1}$) are

$$\hat{P}(s_{t-1}, s_t) = \frac{e^{\beta V(s_t)}}{\sum_{s'} e^{\beta V(s')}}$$ with $\beta$ chosen to normalize the $V(s')$ to have standard deviation 1, as in the previous models.

## Action value model

In this model, we assume that dopamine activity simply reflects the probability of each transition rather than encoding a prediction error; that is, we assume $P(s_{t-1}, s_t) = \frac{D(s_{t-1}, s_t)}{\sum_s D(s_{t-1}, s)}$.

## State value model

In this model, we assume that dopamine activity simply reflects the probability of each behavioral syllable being chosen and is independent of the previous syllable. That is, we compute the average dopamine activity $D(s)$ associated with each syllable $s$, and predict $P(s_{t-1}, s_t) = \frac{D(s_t)}{\sum_s D(s)}$.

# Acknowledgements

We thank Jaeeon Lee for providing the initial inspiration for this project, Sean Escola for fruitful discussions, and Steven A Siegelbaum for comments on the manuscript. JL is supported by the Mathers Foundation and the Gatsby Charitable Foundation. JM is supported by a Career Award at the Scientific Interface from the Burroughs Wellcome Fund, a fellowship from the Sloan Foundation, and a fellowship from the David and Lucille Packard Foundation. SRD is supported by NIH grants RF1AG073625, R01NS114020, and U24NS109520, the Simons Foundation Autism Research Initiative, and the Simons Collaboration on Plasticity and the Aging Brain. ALK is supported by the Mathers Foundation, the Burroughs Wellcome Foundation, the McKnight Endowment Fund, and the Gatsby Charitable Foundation.

# Additional information

### Competing interests

Sandeep R Datta: S.R.D. sits on the scientific advisory boards of Neumora and Gilgamesh Therapeutics, which have licensed or sub-licensed the MoSeq technology. The other authors declare that no competing interests exist.

### Funding

| Funder | Grant reference number | Author |
| --- | --- | --- |
| Mathers Foundation | | Jack W Lindsey<br>Ashok Litwin-Kumar |
| Gatsby Charitable Foundation | | Jack W Lindsey<br>Ashok Litwin-Kumar |
| Burroughs Wellcome Fund | Career Award at the Scientific Interface | Jeffrey Markowitz<br>Ashok Litwin-Kumar |
| Sloan Foundation | | Jeffrey Markowitz |
| David and Lucile Packard Foundation | | Jeffrey Markowitz |
| National Institutes of Health | RF1AG073625 | Sandeep R Datta |
| National Institutes of Health | R01NS114020 | Sandeep R Datta |
| National Institutes of Health | U24NS109520 | Sandeep R Datta |
| Simons Foundation Autism Research Initiative | | Sandeep R Datta |

| Funder | Grant reference number | Author |
|---|---|---|
| McKnight Endowment Fund for Neuroscience | | Ashok Litwin-Kumar |

The funders had no role in study design, data collection, and interpretation, or the decision to submit the work for publication.

### Author contributions
Jack W Lindsey, Conceptualization, Investigation, Writing – original draft, Writing – review and editing; Jeffrey Markowitz, Winthrop F Gillis, Sandeep R Datta, Ashok Litwin-Kumar, Conceptualization, Writing – original draft, Writing – review and editing

### Author ORCIDs
Jack W Lindsey (ID) https://orcid.org/0000-0003-0930-7327
Jeffrey Markowitz (ID) https://orcid.org/0000-0003-2362-1937
Ashok Litwin-Kumar (ID) https://orcid.org/0000-0003-2422-6576

Reviewer #1 (Public review): https://doi.org/10.7554/eLife.101747.3.sa1
Reviewer #2 (Public review): https://doi.org/10.7554/eLife.101747.3.sa2
Reviewer #3 (Public review): https://doi.org/10.7554/eLife.101747.3.sa3
Author response https://doi.org/10.7554/eLife.101747.3.sa4

## Additional files

### Supplementary files
MDAR checklist

### Data availability
Code implementing the model is available on GitHub (https://github.com/alitwinkumar/lindsey_etal_striatal_dynamics, copy archived at *Litwin-Kumar, 2025*).

The following previously published dataset was used:

| Author(s) | Year | Dataset title | Dataset URL | Database and Identifier |
|---|---|---|---|---|
| Markowitz JE, Gillis WF, Jay M, Wood J, Harris R, Cieszkowski R, Scott R, Brann D, Koveal D, Kula T, Weinreb C, Osman MA, Pinto SR, Uchida N, Linderman S, Sabatini B, Datta SR | 2023 | Spontaneous behaviour is structured by reinforcement without explicit reward | https://doi.org/10.5281/zenodo.7274803 | Zenodo, 10.5281/zenodo.7274803 |

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

## Appendix 1

### Relationship between sum mode activity and future difference mode activity

In the main text, we provided an argument for why sum mode activity drives changes to future difference mode activity, assuming a linear $f^{\text{d/iSPN}}(\delta)$ and linear neural activation functions. Here, we generalize this argument to more general learning rules and activation functions $\phi$, assuming only that $f^{\text{dSPN}}(\delta)$ is monotonically increasing, $f^{\text{iSPN}}(\delta)$ is monotonically increasing, and $\phi(\cdot)$ is monotonically increasing. We have that $y^{\text{d/iSPN}} = \phi(\mathbf{w}^{\text{d/iSPN}} \cdot \mathbf{x})$, and $\delta\mathbf{w}^{\text{d/iSPN}} = (f^{\text{d/iSPN}}(\delta) \cdot y^{\text{d/iSPN}})\mathbf{x}$. Thus, in the limit of small small weight updates, we can write:

$$
\begin{aligned}
\Delta(y^{\text{dSPN}} - y^{\text{iSPN}}) &= \Delta\phi(\mathbf{w}^{\text{dSPN}} \cdot \mathbf{x}) - \Delta\phi(\mathbf{w}^{\text{iSPN}} \cdot \mathbf{x}) \\
&\approx \phi'(\mathbf{w}^{\text{dSPN}} \cdot \mathbf{x})(\Delta\mathbf{w}^{\text{dSPN}} \cdot \mathbf{x}) - \phi'(\mathbf{w}^{\text{iSPN}} \cdot \mathbf{x})(\Delta\mathbf{w}^{\text{iSPN}} \cdot \mathbf{x}) \\
&\propto \phi'(\mathbf{w}^{\text{dSPN}} \cdot \mathbf{x})(f^{\text{dSPN}}(\delta) \cdot y^{\text{dSPN}}\mathbf{x} \cdot \mathbf{x}) - \phi'(\mathbf{w}^{\text{iSPN}} \cdot \mathbf{x})(f^{\text{iSPN}}(\delta) \cdot y^{\text{iSPN}}\mathbf{x} \cdot \mathbf{x}) \\
&= \|x\|^2 \left( \phi'(\mathbf{w}^{\text{dSPN}} \cdot \mathbf{x})(f^{\text{dSPN}}(\delta) \cdot y^{\text{dSPN}}) - \phi'(\mathbf{w}^{\text{iSPN}} \cdot \mathbf{x})(f^{\text{iSPN}}(\delta) \cdot y^{\text{iSPN}}) \right) \\
&\propto c^{\text{dSPN}}f^{\text{dSPN}}(\delta)y^{\text{dSPN}} + (-c^{\text{iSPN}}f^{\text{iSPN}}(\delta)y^{\text{iSPN}}),
\end{aligned}
\tag{24}
$$

where $c^{\text{dSPN}}$ and $c^{\text{iSPN}}$ are nonnegative because $\phi'$ is always nonnegative by assumption. Since by assumption $f^{\text{d/iSPN}}$ are increasing/decreasing, respectively, the first term of the above sum has nonnegative correlation with $\delta y^{\text{dSPN}}$ and the second term has nonnegative correlation with $\delta y^{\text{iSPN}}$. Thus, changes $\Delta(y^{\text{dSPN}} - y^{\text{iSPN}})$ to difference mode activity are always nonnegatively correlated with sum mode activity. If we assume that efferent excitation is always sufficiently strong that $c^{\text{dSPN}} = \phi'(\mathbf{w}^{\text{dSPN}} \cdot \mathbf{x})$ and $c^{\text{iSPN}} = \phi'(\mathbf{w}^{\text{iSPN}} \cdot \mathbf{x})$ are positive, and that there are no values of $\delta$ for which $f^{\text{d/iSPN}}(\delta)$ both have zero derivative, we can further guarantee that changes to difference mode activity will always be *positively* correlated with sum mode activity.

### Generalizing the model to a distributed code for actions

In our model simulations in the main text, we assumed for convenience that there is a single dSPN and iSPN that promote and suppress each available action, respectively. It is more realistic to model the code for action as distributed among many SPNs. Our model generalizes easily to this case; all that is necessary is for the efferent activity following action selection to excite the vectors (for both dSPNs and iSPNs) in population activity space corresponding to that action. To demonstrate this, we conducted a simulation with $N = 1000$ dSPNs and iSPNs each, $S = 10$ input cues (one-hot input vectors), and $A = 10$ actions, with one correct action for each input state. Feedforward SPN activity is given by

$$
y_i^{\text{dSPN}} = \phi\left( \sum_{j=1}^{M} w_{ij}^{\text{dSPN}} x_j \right),
\tag{25}
$$

$$
y_i^{\text{iSPN}} = \phi\left( \sum_{j=1}^{M} w_{ij}^{\text{iSPN}} x_j \right).
\tag{26}
$$

The log-likelihood of an action $a$ being performed is proportional to

$$
\ell_a = \sum_{i=1}^{N} \zeta_{ai}^{\text{dSPN}} y_i^{\text{dSPN}} - \zeta_{ai}^{\text{iSPN}} y_i^{\text{iSPN}},
\tag{27}
$$

where $\zeta_{ai}^{\text{dSPN}}$ and $\zeta_{ai}^{\text{iSPN}}$ are randomly sampled uniformly in the interval $[0, 1]$ and then normalized so that each vector $\zeta_a^{\text{dSPN}}$ and $\zeta_a^{\text{iSPN}}$ has norm 1. Thus, the contribution of each dSPN/iSPN to the promotion/suppression of each action is randomly distributed.

In the efference model, following selection of an action $a^*$, activities of the SPNs associated with action $a^*$ are updated as follows, so that efference excites the modes $\zeta_{a^*}^{\text{dSPN}}$ and $\zeta_{a^*}^{\text{iSPN}}$ associated with the selected action:

$$y_i^{\text{dSPN}} \leftarrow \phi \left( c_{\text{efference}} \cdot \zeta_{a^*i}^{\text{dSPN}} + \sum_{j=1}^{M} w_{ij}^{\text{dSPN}} x_j \right), \tag{28}$$

$$y_i^{\text{iSPN}} \leftarrow \phi \left( c_{\text{efference}} \cdot \zeta_{a^*i}^{\text{iSPN}} + \sum_{j=1}^{M} w_{ij}^{\text{iSPN}} x_j \right). \tag{29}$$

We also experiment with a generalization of the canonical action selection model to this distributed action tuning architecture, in which following action selection, SPN activity is set to

$$y_i^{\text{dSPN}} \leftarrow \zeta_{a^*i}^{\text{dSPN}}, \tag{30}$$

$$y_i^{\text{iSPN}} \leftarrow \left( \max_{i'} \zeta_{a^*i'}^{\text{iSPN}} \right) - \zeta_{a^*i}^{\text{iSPN}}. \tag{31}$$

In this model, dSPNs are excited in proportion to their contribution to the currently selected action and iSPNs are suppressed in proportion to their degree of inhibition of the currently selected action.

The plasticity rules used are the same as in the main text.

We find that the results of the main text – that the canonical action selection model fails to learn from negative rewards, while the efference model successfully learns from both reward protocols – is replicated (*Figure 3—figure supplement 1*).

## Alternative model with shared plasticity rule among all SPNs

The issues identified in *Figure 2* with the canonical action selection model are a consequence of the iSPN plasticity rule. From a normative perspective, it is interesting to consider why the empirically observed iSPN plasticity rule might be advantageous, compared to an alternative model in which iSPNs share the same plasticity rule as dSPNs. For instance, this alternative model can solve the two-alternative forced choice task of *Figure 2* with both positive and negative reward protocols (*Figure 7—figure supplement 1*, left). However, the limitations of this alternative model are revealed in the off-policy learning setting, where the Q-learning algorithm is required. In this case, SPN activity must encode Q-values associated with each action, but in the canonical action selection model, these values are disrupted by the updates to SPN activity following action selection. This is because the activity updates in the canonical action selection model modify difference mode activity, which (when dSPN and iSPN plasticity rules are the same) is needed for learning (*Figure 7—figure supplement 1B*). As a result, the predicted Q-values are inaccurate, and the model has difficulty learning the true value of each action. We demonstrate this in the two-alternative forced task in an off-policy learning protocol where an oracle chooses the correct action on each trial, and the striatal pathway's ability to solve the task independently is evaluated. The efference activity model has no issue due to the orthogonality of the efferent activity and difference modes as described above, but the canonical action selection model fails to solve the task (*Figure 7—figure supplement 1A*, right).

We note that non-orthogonality of the activity mode used for learning and behavior could cause other problems besides impairing the system's ability to implement off-policy learning algorithms; for instance, even in an on-policy setting, it could interfere with sequential action selection at short timescales.

