## [Editor Report · eLife Assessment]

The authors present a biologically plausible framework for action selection and learning in the striatum that is a **fundamental** advance in our understanding of possible neural implementations of reinforcement learning in the basal ganglia. They provide **compelling** evidence that their model can reconcile realistic neural plasticity rules with the distinct functional roles of the direct and indirect spiny projection neurons of the striatum, recapitulating experimental findings regarding the activity profiles of these distinct neural populations and explaining a key aspect of striatal function.

---

## [Referee Report · Reviewer #1 (Public review)]

Summary:

The authors propose a new model of biologically realistic reinforcement learning in the direct and indirect pathway spiny projection neurons in the striatum. These pathways are widely considered to provide a neural substrate for reinforcement learning in the brain. However, we do not yet have a full understanding of mechanistic learning rules that would allow successful reinforcement learning like computations in these circuits. The authors outline some key limitations of current models and propose an interesting solution by leveraging learning with efferent inputs of selected actions. They show that the model simulations are able to recapitulate experimental findings about the activity profile in these populations in mice during spontaneous behavior. They also show how their model is able to implement off-policy reinforcement learning.

Strengths:

The manuscript has been very clearly written and the results have been presented in a readily digestible manner. The limitations of existing models, that motive the presented work, have been clearly presented and the proposed solution seems very interesting. The novel contribution in the proposed model is the idea that different patterns of activity drive current action selection and learning. Not only does this allow the model is able to implement reinforcement learning computations well, this suggestion may have interesting implications regarding why some processes selectively affect ongoing behavior and others affect learning. The model is able to recapitulate some interesting experimental findings about various activity characteristics of dSPN and iSPN pathway neuronal populations in spontaneously behaving mice. The authors also show that their proposed model can implement off-policy reinforcement learning algorithms with biologically realistic learning rules. This is interesting since off-policy learning provides some unique computational benefits and it is very likely that learning in neural circuits may, at least to some extent, implement such computations.

Weaknesses:

A weakness in this work is that it isn't clear how a key component in the model - an efferent copy of selected actions - would be accessible to these striatal populations. The authors propose several plausible candidates, but future work may clarify the feasibility of this proposal.

---

## [Referee Report · Reviewer #2 (Public review)]

Summary:

The basal ganglia is often understood within a reinforcement learning (RL) framework, where dopamine neurons convey a reward prediction error which modulates cortico-striatal connections onto spiny projection neurons (SPNS) in the striatum. However, current models of plasticity rules are inconsistent with learning in a reinforcement learning framework.

This paper proposes a new model that describes how distinct learning rules in direct and indirect pathway striatal neurons allows them to implement reinforcement learning models. It proposes that two distinct component of striatal activity affect action selection and learning. They show that the proposed implementation allows learning in simple tasks and is consistent with experimental data from calcium imaging data in direct and indirect SPNs in freely moving mouse.

Strengths:

Despite the success of reward prediction errors at characterizing the responses of dopamine neurons as the temporal difference error within an RL framework, the implementation of RL algorithms in the rest of the basal ganglia has been unclear. A key missing aspect has been the lack of a RL implementation that is consistent with the distinction of direct- and indirect SPNs. This paper proposes a new model that is able to learn successfully in simple RL tasks and explains recent experimental results.

The author shows that their proposed model, unlike previous implementations, this model can perform well in RL tasks. The new model allows them to make experimental predictions. They test some of these predictions and show that the dynamics of dSPNs and iSPNs correspond to model predictions.

More generally, this new model can be used to understand striatal dynamics across direct and indirect SPNs in future experiments.

Weaknesses:

The authors could characterize better the reliability of their experimental predictions and the description of the parameters of some of the simulations

The authors propose some ideas about how the specificity of the striatal efferent inputs but should highlight better that this is a key feature of the model whose anatomical implementation has yet to be resolved.

Comments on revisions:

I thank the authors for their response to public and private reviews and for the clarifications and changes to the manuscript which have strengthened it. I understand the inability to implement some of the proposed additional simulation due to authors having left academia and the request for a version of record.

---

## [Referee Report · Reviewer #3 (Public review)]

Summary:

This paper points out an inconsistency of the roles of the striatal spiny neurons projecting to the indirect pathway (iSPN) and the synaptic plasticity rule of those neurons expressing dopamine D2 receptors, and proposes a novel, intriguing mechanisms that iSPNs are activated by the efference copy of the chosen action that they are supposed to inhibit.

The proposed model was supported by simulations and analysis of the neural recording data during spontaneous behaviors.

Strengths:

Previous models suggested that the striatal neurons learn action values functions, but how the information about the chosen action is fed back to the striatum for learning was not clear. The author pointed out that this is a fundamental problem for iSPNs that are supposed to inhibit specific actions and its synaptic inputs are potentiated with dopamine dips.

The authors proposes a novel hypothesis that iSPNs are activated by efference copy of the selected action which they are supposed to inhibit during action selection. Even though intriguing and seemingly unnatural, the authors demonstrated that the model based on the hypothesis can circumvent the problem of iSPNs learning to disinhibit the actions associated with negative reward errors. They further showed by analyzing the cell-type specific neural recording data by Markowitz et al. (2018) that iSPN activities tend to be anti-correlated before and after action selection.

Weaknesses:

(1) It is not correct to call the action value learning using the externally-selected action as "off-policy." Both off-policy algorithm Q-learning and on-policy algorithm SARSA update the action value of the chosen action, which can be different from the greedy action implicated by the present action values. In standard reinforce learning terminology, on-policy or off-policy is regarding the actions in the subsequent state, whether to use the next action value of (to be) chosen action or that of greedy choice as in equation (7).

It is worth noting that this paper suggested that dopamine neurons encode on-policy TD errors: Morris G, Nevet A, Arkadir D, Vaadia E, Bergman H (2006). Midbrain dopamine neurons encode decisions for future action. Nat Neurosci, 9, 1057-63. https://doi.org/10.1038/nn1743

(2) It is also confusing to contract TD learning and Q-learning, as the latter is considered as on type of TD learning. In the TD error signal by state value function (6) is dependent on the chosen action a_{t-1} implicitly in r_t and s_t based on the reward and state transition function.

(3) It is not clear why interferences of the activities for action selection and learning can be avoided, especially when actions are taken with short intervals or even temporal overlaps. How can the efference copy activation for the previous action be dissociated with the sensory cued activation for the next action selection?

(4) Although it may be difficult to single out the neural pathway that carries the efference copy signal to the striatum, it is desired to consider their requirements and difference possibilities. A major issue is that the time delay from actions to reward feedback can be highly variable.

An interesting candidate is the long-latency neurons in the CM thalamus projecting to striatal cholinergic interneurons, which are activated following low-reward actions:

Minamimoto T, Hori Y, Kimura M (2005). Complementary process to response bias in the centromedian nucleus of the thalamus. Science, 308, 1798-801. https://doi.org/10.1126/science.1109154

(5) In the paragraph before Eq. (3), Eq (1) should be Eq. (2) for the iSPN.

Here are comments back to the authors' replies with the revised version:

(1) I do not agree on the use of inaccurate technical terms. On-policy does not require that the policy is greedy with respect to the actions values, as authors seem to assume here.

In fact, the policy (10) is just a standard soft-max action selection based on the action values by the difference of dSPN and iSPN outputs.

Furthermore, in the immediate reward setting tested in this paper, action values are independent of the policy, so there is no distinction between on-policy vs. off-policy. This is also apparent from the "TD" errors in (19) and (21), where there is no TD.

(2) To really compare the different forms of TD, multi-step RL tasks should be used.

(3) This fundamental limitation should be explicitly documented in the manuscript. This is not just the same as any RL algorithms. Having two action representations within each action step make temporal credit assignment more difficult.

---

## [Author Response]

The following is the authors’ response to the original reviews

**Reviewer #1:**
Summary:The authors propose a new model of biologically realistic reinforcement learning in the direct and indirect pathway spiny projection neurons in the striatum. These pathways are widely considered to provide a neural substrate for reinforcement learning in the brain. However, we do not yet have a full understanding of mechanistic learning rules that would allow successful reinforcement learning like computations in these circuits. The authors outline some key limitations of current models and propose an interesting solution by leveraging learning with efferent inputs of selected actions. They show that the model simulations are able to recapitulate experimental findings about the activity profile in these populations of mice during spontaneous behavior. They also show how their model is able to implement off-policy reinforcement learning.Strengths:The manuscript has been very clearly written and the results have been presented in a readily digestible manner. The limitations of existing models, that motivate the presented work, have been clearly presented and the proposed solution seems very interesting. The novel contribution of the proposed model is the idea that different patterns of activity drive current action selection and learning. Not only does this allow the model is able to implement reinforcement learning computations well, but this suggestion may have interesting implications regarding why some processes selectively affect ongoing behavior and others affect learning. The model is able to recapitulate some interesting experimental findings about various activity characteristics of dSPN and iSPN pathway neuronal populations in spontaneously behaving mice. The authors also show that their proposed model can implement off-policy reinforcement learning algorithms with biologically realistic learning rules. This is interesting since off-policy learning provides some unique computational benefits and it is very likely that learning in neural circuits may, at least to some extent, implement such computations.

We thank the reviewer for the positive comments.

Weaknesses:A weakness in this work is that it isn’t clear how a key component in the model - an efferent copy of selected actions - would be accessible to these striatal populations. The authors propose several plausible candidates, but future work may clarify the feasibility of this proposal.

We agree that the biological substrate of the efference copy remains a key open question. We discuss potential pathways in the Discussion section of our manuscript and hope that future experimental studies clarify the question.

**Reviewer #2:**
Summary:The basal ganglia is often understood within a reinforcement learning (RL) framework, where dopamine neurons convey a reward prediction error that modulates cortico-striatal connections onto spiny projection neurons (SPNS) in the striatum. However, current models of plasticity rules are inconsistent with learning in a reinforcement learning framework.This paper proposes a new model that describes how distinct learning rules in direct and indirect pathway striatal neurons allow them to implement reinforcement learning models. It proposes that two distinct components of striatal activity affect action selection and learning. They show that the proposed implementation allows learning in simple tasks and is consistent with experimental data from calcium imaging data in direct and indirect SPNs in freely moving mice.Strengths:Despite the success of reward prediction errors at characterizing the responses of dopamine neurons as the temporal difference error within an RL framework, the implementation of RL algorithms in the rest of the basal ganglia has been unclear. A key missing aspect has been the lack of a RL implementation that is consistent with the distinction of direct- and indirect SPNs. This paper proposes a new model that is able to learn successfully in simple RL tasks and explains recent experimental results.The author shows that their proposed model, unlike previous implementations, this model can perform well in RL tasks. The new model allows them to make experimental predictions. They test some of these predictions and show that the dynamics of dSPNs and iSPNs correspond to model predictions.More generally, this new model can be used to understand striatal dynamics across direct and indirect SPNs in future experiments.

We thank the reviewer for the positive comments.

Weaknesses:The authors could characterize better the reliability of their experimental predictions and the description of the parameters of some of the simulations.

In addition to the descriptions in the Methods, we have provided code implementing the key features of our simulations, which should contribute to reproducibility of our results.

The authors propose some ideas about how the specificity of the striatal efferent inputs but should highlight better that this is a key feature of the model whose anatomical implementation has yet to be resolved.

We have clarified in the Discussion section “Biological substrates of striatal efferent inputs” that these represent assumptions or predictions that have not yet been demonstrated experimentally.

**Reviewer #3:**
Summary:This paper points out an inconsistency of the roles of the striatal spiny neurons projecting to the indirect pathway (iSPN) and the synaptic plasticity rule of those neurons expressing dopamine D2 receptors and proposes a novel, intriguing mechanisms that iSPNs are activated by the efference copy of the chosen action that they are supposed to inhibit.The proposed model was supported by simulations and analysis of the neural recording data during spontaneous behaviors.Strengths:Previous models suggested that the striatal neurons learn action-value functions, but how the information about the chosen action is fed back to the striatum for learning was not clear. The author pointed out that this is a fundamental problem for iSPNs that are supposed to inhibit specific actions and its synaptic inputs are potentiated with dopamine dips.The authors propose a novel hypothesis that iSPNs are activated by efference copy of the selected action which they are supposed to inhibit during action selection. Even though intriguing and seemingly unnatural, the authors demonstrated that the model based on the hypothesis can circumvent the problem of iSPNs learning to disinhibit the actions associated with negative reward errors. They further showed by analyzing the cell-type specific neural recording data by Markowitz et al. (2018) that iSPN activities tend to be anti-correlated before and after action selection.

We thank the reviewer for the positive comments.

Weaknesses:It is not correct to call the action value learning using the externally-selected action as “offpolicy.” Both off-policy algorithm Q-learning and on-policy algorithm SARSA update the action value of the chosen action, which can be different from the greedy action implicated by the present action values. In standard reinforcement learning terminology, on-policy or off-policy is regarding the actions in the subsequent state, whether to use the next action value of (to be) chosen action or that of greedy choice as in equation (7).It is worth noting that this paper suggested that dopamine neurons encode on-policy TD errors: Morris G, Nevet A, Arkadir D, Vaadia E, Bergman H (2006). Midbrain dopamine neurons encode decisions for future action. Nat Neurosci, 9, 1057-63. https://doi.org/10.1038/nn1743.

We regret that we do not completely follow the reviewer’s comment. We use “off-policy” to refer to the fact that, considered in isolation, the basal ganglia reinforcement learning system that we model learns a target policy that may be distinct from the behavioral policy of the organism as a whole.

It is also confusing to contract TD learning and Q-learning, as the latter is considered as one type of TD learning. In the TD error signal by state value function (6) is dependent on the chosen action *at*−1 implicitly in *rt* and *st* based on the reward and state transition function.

We agree that this was confusing. We have therefore changed the places in our paper where we intended to refer to “TD learning of a value function *V* (*s*)” to specifically mention *V* (*s*), rather than just “TD learning.”

It is not clear why interferences of the activities for action selection and learning can be avoided, especially when actions are taken with short intervals or even temporal overlaps. How can the efference copy activation for the previous action be dissociated with the sensory cued activation for the next action selection?

The non-interference arises from the orthogonality of the difference (action selection) and sum (efference copy) modes, as described in Figure 3. However, we agree with the reviewer that the problem of temporal credit assignment, when many actions are taken before reward feedback is obtained, is present in our model, as in any standard RL model.

Although it may be difficult to single out the neural pathway that carries the efference copy signal to the striatum, it is desired to consider their requirements and difference possibilities. A major issue is that the time delay from actions to reward feedback can be highly variable.An interesting candidate is the long-latency neurons in the CM thalamus projecting to striatal cholinergic interneurons, which are activated following low-reward actions: Minamimoto T, Hori Y, Kimura M (2005). Complementary process to response bias in the centromedian nucleus of the thalamus. Science, 308, 1798-801. https://doi.org/10.1126/science.1109154.

We are grateful for the interesting suggestion and reference, which we have added to the manuscript. However, we note that the issue of delayed reward feedback may also be partially addressed by using a sufficiently long eligibility trace.

In the paragraph before Eq. (3), Eq. (1) should be Eq. (2) for the iSPN.

Corrected.